# Differentially Private Sampling from Distributions

**Sofya Raskhodnikova**
Department of Computer Science
Boston University
sofya@bu.edu

**Satchit Sivakumar**
Department of Computer Science
Boston University
satchit@bu.edu

**Adam Smith**
Department of Computer Science
Boston University
ads22@bu.edu

**Marika Swanberg**
Department of Computer Science
Boston University
marikas@bu.edu

## Abstract

We initiate an investigation of private sampling from distributions. Given a dataset with $n$ independent observations from an unknown distribution $P$, a sampling algorithm must output a single observation from a distribution that is close in total variation distance to $P$ while satisfying differential privacy. Sampling abstracts the goal of generating small amounts of realistic-looking data. We provide tight upper and lower bounds for the dataset size needed for this task for three natural families of distributions: arbitrary distributions on $\{1, \dots, k\}$, arbitrary product distributions on $\{0, 1\}^d$, and product distributions on on $\{0, 1\}^d$ with bias in each coordinate bounded away from 0 and 1. We demonstrate that, in some parameter regimes, private sampling requires asymptotically fewer observations than learning a description of $P$ nonprivately; in other regimes, however, private sampling proves to be as difficult as private learning. Notably, for some classes of distributions, the overhead in the number of observations needed for private learning compared to non-private learning is completely captured by the number of observations needed for private sampling.

## 1   Introduction

Statistical machine learning models trained on sensitive data are now widely deployed in domains such as education, finance, criminal justice, medicine, and public health. The personal data used to train such models are more detailed than ever, and there is a growing awareness of the privacy risks that come with their use. Differential privacy is a standard for confidentiality that is now well studied and increasingly deployed.

Differentially private algorithms ensure that whatever is learned about an individual from the algorithm's output would be roughly the same whether or not that individual's record was actually part of the input dataset. This requirement entails a strong guarantee, but limits the design of algorithms significantly. As a result, there is a substantial line of work on developing good differentially private methodology for learning and statistical estimation tasks, and on understanding how the sample size required for specific tasks increases relative to unconstrained algorithms. A typical task investigated in this area is *distribution learning*: informally, given records drawn i.i.d. from an unknown distribution $P$, the algorithm aims to produce a description of a distribution $Q$ that is close to $P$.

However, often the task at hand requires much less than full-fledged learning. We may simply need to generate a small amount of data that has the same distributional properties as the population, or perhaps simply "looks plausible" for the population. For example, one might need realistic data for debugging a software program or for getting a quick idea of the range of values in a dataset.

35th Conference on Neural Information Processing Systems (NeurIPS 2021).

In this work, we study a basic problem that captures these seemingly less stringent aims. Informally, the goal is to design a sampling algorithm that, given a dataset with $n$ observations drawn i.i.d. from a distribution $P$, generates a single observation from a distribution $Q$ that is close to $P$.

To formulate the problem precisely, consider a randomized algorithm $\mathcal{A} : \mathcal{U}^n \to \mathcal{U}$ that takes as input a dataset of $n$ records from some universe $\mathcal{U}$ and outputs a single element of $\mathcal{U}$. Given a distribution $P$ on $\mathcal{U}$, let $\mathbf{X} = (X_1, ... X_n)$ be a random dataset with entries drawn i.i.d. from $P$. Let $\mathcal{A}(\mathbf{X})$ denote the random variable corresponding to the output of the algorithm $\mathcal{A}$ on input $\mathbf{X}$. This random variable depends on both the selection of entries of $\mathbf{X}$ from $P$ and the coins of $\mathcal{A}$. Let $Q_{\mathcal{A},P}$ denote the distribution of $\mathcal{A}(\mathbf{X})$, so that

$$Q_{\mathcal{A},P}(z) = \Pr_{\substack{\mathbf{X} \sim_{i.i.d.} P \\ \text{coins of } \mathcal{A}}} (\mathcal{A}(\mathbf{X}) = z) = \sum_{\mathbf{x} \in \mathcal{U}^n} \left( \Pr(\mathbf{X} = \mathbf{x}) \cdot \Pr_{\text{coins of } \mathcal{A}} (\mathcal{A}(\mathbf{x}) = z) \right).$$

The dataset size $n$ is a parameter of $\mathcal{A}$ and thus defined implicitly. We measure the closeness between the input and output distributions in total variation distance, denoted $d_{TV}$.

**Definition 1.1** (Accuracy of sampling [4]). *A sampler $\mathcal{A}$ is $\alpha$-accurate on a distribution $P$ if*

$$d_{TV}(Q_{\mathcal{A},P}, P) \leq \alpha.$$

*A sampler is $\alpha$-accurate on a class $\mathcal{C}$ of distributions if it is $\alpha$-accurate on every $P$ in $\mathcal{C}$.*

The class $\mathcal{C}$ in the accuracy definition above effectively encodes what the sampler is allowed to assume about $P$. For example, $\mathcal{C}$ might include all distributions on $k$ elements or all product distributions on $\{0, 1\}^d$ (that is, distributions on $d$-bit strings for which the individual binary entries are independent).

Our aim in formulating Definition 1.1 was to capture the weakest reasonable task that abstracts the goal of generating data with the same distributional properties as the input data. Without any privacy constraints, this task is trivial to achieve: an algorithm that simply outputs its first input record will sample from exactly the distribution $P$, so the interesting problem is to generate a sample of size $m > n$ when given only $n$ observations (as in the work of Axelrod et al. [4], which inspired our investigation). However, a differentially private algorithm cannot, in general, simply output one of its records in the clear. Even producing a single correctly distributed sample is a non-trivial task.

To understand the task and compare our results to existing work, it is helpful to contrast our definition of sampling with the more stringent goal of learning a distribution. A learning algorithm gets input in the same format as a sampling algorithm. We state a definition of distribution learning formulated with a single parameter $\alpha$ that captures both the distance between distributions and the failure probability.

**Definition 1.2** (Distribution learning). *An algorithm $\mathcal{B}$ learns a distribution class $\mathcal{C}$ to within error $\alpha$ if, given a dataset consisting of independent observations from a distribution $P \in \mathcal{C}$, algorithm $\mathcal{B}$ outputs a description of a distribution that satisfies*

$$\Pr_{\substack{\mathbf{X} \sim_{i.i.d.} P \\ \text{coins of } \mathcal{B}}} \left( d_{TV}(\mathcal{B}(\mathbf{X}), P) \leq \alpha \right) \geq 1 - \alpha.$$

An algorithm $\mathcal{B}$ that learns class $\mathcal{C}$ to within error $\alpha$ immediately yields a $2\alpha$-accurate sampler for class $\mathcal{C}$: the sampler first runs $\mathcal{B}$ to get a distribution $\hat{P}$ and then generates a single sample from $\hat{P}$. Thus, it is instructive to compare bounds on the dataset size required for sampling to known results on the sample complexity of distribution learning. Lower bounds for sampling imply lower bounds for all more stringent tasks, including learning, whereas separations between the complexity of sampling and that of learning suggest settings where weakening the learning objective might be productive.

**Differential privacy**    Differential privacy is a constraint on the algorithm $\mathcal{A}$ that processes a dataset. It does not rely on any distributional property of the data. We generally use uppercase letters (e.g., $\mathbf{X} = (X_1, ..., X_n)$) when viewing the data as a random variable, and lowercase symbols (e.g., $\mathbf{x} = (x_1, ..., x_n)$) when treating the data as fixed. Two datasets $\mathbf{x}, \mathbf{x}' \in \mathcal{U}^n$ are *neighbors* if they differ in at most one entry. If each entry corresponds to the data of one person, then neighboring datasets differ by replacing one person's data with an alternate record. Informally, differential privacy requires that an algorithm's output distributions are similar on all pairs of neighboring datasets.

**Definition 1.3** (Differential Privacy [15, 14]). *A randomized algorithm $\mathcal{A} : \mathcal{U}^n \to \mathcal{Y}$ is $(\varepsilon, \delta)$-differentially private (in short, $(\varepsilon, \delta)$-DP) if for every pair of neighboring datasets $\mathbf{x}, \mathbf{x}' \in \mathcal{U}^n$ and for all subsets $Y \subseteq \mathcal{Y}$,*

$$\Pr[\mathcal{A}(\mathbf{x}) \in Y] \leq e^\varepsilon \cdot \Pr[\mathcal{A}(\mathbf{x}') \in Y] + \delta.$$

Table 1: Sample complexity of sampling and estimation tasks. Our negative results hold for $(\varepsilon, \delta)$-differential privacy when $\delta < 1/n$. In this table, $\varepsilon \leq 1$ and $\delta = 1/n^c$ for constant $c > 1$.

| | Distributions on $[k]$ | Product distributions on $\{0,1\}^d$ | Product distributions with $p_j \in [\frac{1}{3}, \frac{2}{3}]$ |
|---|---|---|---|
| Nonprivate learning | $\Theta\left(\dfrac{k}{\alpha^2}\right)$ | $\Theta\left(\dfrac{d}{\alpha^2}\right)$ | |
| $(\varepsilon, \delta)$-DP sampling (this work) | $\Theta\left(\dfrac{k}{\alpha\varepsilon}\right)$ 

 Theorems 1.4, 1.5 | $\tilde{O}\left(\dfrac{d}{\alpha\varepsilon}\right)$ 

 Theorem 1.6 | $\tilde{O}\left(\dfrac{\sqrt{d}}{\varepsilon} + \log\dfrac{d}{\alpha}\right)$ 

 Theorem 1.8 |
| | | $\Omega(d)$    Theorem 1.7 | $\Omega\left(\sqrt{d}/\varepsilon\right)$    Theorem 1.9 |
| $(\varepsilon, \delta)$-DP learning | $\Theta\left(\dfrac{k}{\alpha^2} + \dfrac{k}{\alpha\varepsilon}\right)$   [11] | $\tilde{\Theta}\left(\dfrac{d}{\alpha\varepsilon} + \dfrac{d}{\alpha^2}\right)$ [17, 8] | |

For $\varepsilon \leq 1$ and $\delta < 1/n$, this guarantee implies, roughly, that one can learn the same things about any given individual from the output of the algorithm as one could had their data not been used in the computation [19]. When $\delta = 0$, the guarantee is referred to as pure differential privacy.

## 1.1 Our results

We initiate a systematic investigation of the sample complexity of differentially private sampling. We provide upper and lower bounds for the sample size needed for this task for three natural families of distributions: arbitrary distributions on $[k] = \{1, \ldots, k\}$, arbitrary product distributions on $\{0,1\}^d$, and product distributions on $\{0,1\}^d$ with bounded bias. We demonstrate that, in some parameter regimes, private sampling requires asymptotically fewer samples than learning a description of $P$ nonprivately; in other regimes, however, sampling proves to be as difficult as private learning.

Our results are summarized in Table 1. Proofs of all results are included in the supplementary material. For simplicity, the table and our informal discussions focus on $(\varepsilon, \delta)$-differential privacy (Definition 1.3) in the setting[1] where $\delta = 1/n^c$ for some constant $c > 1$.

Let $\mathcal{C}_k$ be the class of $k$-ary distributions (that is, distributions on $[k]$). We show that $n = \Theta\left(\frac{k}{\alpha\varepsilon}\right)$ observations are necessary and sufficient for differentially private sampling from $\mathcal{C}_k$ in the worst case.

**Theorem 1.4.** *For all $k \geq 2, \varepsilon > 0$, and $\alpha \in (0,1)$, there exists an $(\varepsilon, 0)$-DP sampler that is $\alpha$-accurate on the distribution class $\mathcal{C}_k$ for datasets of size $n = O(\frac{k}{\alpha\varepsilon})$.*

**Theorem 1.5.** *For all $k \geq 2$, $n \in \mathbb{N}$, $\alpha \in (0, \frac{1}{50}]$, $\varepsilon \in (0,1]$, and $\delta \in (0, \frac{1}{5000n}]$, if there is an $(\epsilon, \delta)$-DP sampler that is $\alpha$-accurate on the distribution class $\mathcal{C}_k$ on datasets of size $n$, then $n = \Omega(\frac{k}{\alpha\varepsilon})$.*

The second major class we consider consists of products of $d$ Bernoulli distributions, for $d \in \mathbb{N}$. We denote this class by $\mathcal{B}^{\otimes d}$. Each distribution in $\mathcal{B}^{\otimes d}$ is described by a vector $(p_1, ..., p_d) \in [0,1]^d$ of $d$ probabilities, called *biases*; a single observation in $\{0,1\}^d$ is generated by flipping $d$ independent coins with respective probabilities $p_1, ..., p_d$ of heads. We show that $n = \tilde{\Theta}(d)$ observations are necessary and sufficient for differentially private sampling from $\mathcal{B}^{\otimes d}$ in the worst case.

**Theorem 1.6.** *For all $d, n \in \mathbb{N}$ and $\epsilon, \delta, \alpha \in (0,1)$, there exists an $(\epsilon, \delta)$-DP sampler that is $\alpha$-accurate on the distribution class $\mathcal{B}^{\otimes d}$ for datasets of size $n = \tilde{O}\left(\frac{d}{\alpha\epsilon}\right)$, assuming $\log(1/\delta) = poly(\log n)$.*

**Theorem 1.7.** *For all sufficiently small $\alpha > 0$, and for all $d, n \in \mathbb{N}$, $\varepsilon \in (0,1]$, and $\delta \in \left[0, \frac{1}{5000n}\right]$, if there is an $(\varepsilon, \delta)$-DP sampler that is $\alpha$-accurate on the distribution class $\mathcal{B}^{\otimes d}$ on datasets of size $n$, then $n = \Omega(d)$.*

Finally, we give better samplers and matching lower bounds for Bernoulli distributions and, more generally, products of Bernoulli distributions, with bias bounded away from 0 and 1. For simplicity,

---

[1]This setting precludes trivial solutions (which are possible when $\delta = \Omega(1/n)$), but allows us to treat factors of $\log(1/\delta)$ as logarithmic in $n$ and absorb them in $\tilde{O}$ expressions.

we consider distributions with bias $p_j \in [\frac{1}{3}, \frac{2}{3}]$ in each coordinate $j \in [d]$. For this class, we show that differentially private sampling can be performed with datasets of size roughly $\sqrt{d}/\varepsilon$, significantly smaller than in the general case. Curiously, the accuracy parameter $\alpha$ has almost no effect on the sample complexity. For Bernoulli distributions with bounded bias, we achieve this with pure differential privacy, that is, with $\delta = 0$. For products of Bernoulli distributions, we need $\delta > 0$.

**Theorem 1.8.** *For all $d \in \mathbb{N}$ and $\epsilon, \delta, \alpha \in (0,1)$, there exists an $(\varepsilon, \delta)$-DP sampler that is $\alpha$-accurate on the class of products of $d$ Bernoulli distributions with biases in $\left[\frac{1}{3}, \frac{2}{3}\right]$ for datasets of size*

$$n = O\left(\frac{\sqrt{d \log(1/\delta)}}{\varepsilon} + \log \frac{d}{\alpha}\right). \text{ When } d = 1, \text{ the sampler has } \delta = 0 \text{ and } n = O(\frac{1}{\varepsilon} + \log \frac{1}{\alpha}).$$

**Theorem 1.9.** *For all sufficiently small $\alpha > 0$, and for all $d, n \in \mathbb{N}$, $\varepsilon \in (0,1]$, and $\delta \in [0, \frac{1}{100n}]$, if there exists an $(\varepsilon, \delta)$-DP sampler that is $\alpha$-accurate on the class of products of $d$ Bernoulli distributions with biases in $\left[\frac{1}{3}, \frac{2}{3}\right]$ on datasets of size $n$, then $n = \Omega(\sqrt{d}/\varepsilon)$.*

**Implications**   Our results show that the sample complexity of private sampling can differ substantially from that of private learning (for which known bounds are stated in Table 1). In some settings, sampling is much easier than learning: for example, for products of Bernoulli distributions with bounded biases, private sampling has a lower dependence on the dimension (specifically, $\sqrt{d}$ instead of $d$) and essentially no dependence on $\alpha$. Even for arbitrary biases or arbitrary $k$-ary distributions, private sampling is easier when $\alpha \ll \varepsilon$. In other settings, however, private sampling can be as hard as private learning: e.g., for $\epsilon \leq \alpha$, the worst-case complexity of sampling and learning $k$-ary distributions and product distributions is the same.

A more subtle point is that, in settings where private sampling is as hard as private learning, sampling accounts for the entire cost of privacy in learning. Specifically, the optimal sample complexity of differentially private learning for arbitrary $k$-ary distributions is $n = \Theta\left(\frac{k}{\alpha^2} + \frac{k}{\alpha\varepsilon}\right)$ (e.g., see [3, Theorem 13]). This bound is the sum of two terms: the sample complexity of nonprivate learning ($\Theta\left(\frac{k}{\alpha^2}\right)$) plus a term to account for the privacy constraint ($\Theta\left(\frac{k}{\alpha\varepsilon}\right)$). One interpretation of our result that private sampling requires $n = \Theta\left(\frac{k}{\alpha\varepsilon}\right)$ observations is that *the extra privacy term in the complexity of learning can be explained by the complexity of privately generating a single sample with approximately correct distribution.*

Another implication of our results is that, *in some settings, the distributions that are hardest for learning—nonprivate or private—are the easiest for sampling, and vice versa.* Consider the simple case of Bernoulli distributions (i.e., product distributions with $d = 1$). The "hard" instances for nonprivate learning to within error $\alpha$ are distributions with bias $p = \frac{1 \pm \alpha}{2}$, but private sampling is easiest in that parameter regime. In contrast, the "hard" instances in our $\Omega(\frac{1}{\alpha\varepsilon})$ lower bound for Bernoulli distributions have bias $10\alpha$, that is, close to 0 as opposed to close to 1/2. A simple variance argument shows that nonprivate learning is easy in that parameter regime, requiring only $O(\frac{1}{\alpha})$ observations. Similarly, for product distributions, we show that the complexity of private sampling is only $\tilde{\Theta}(\sqrt{d})$ when biases are bounded away from 0 and 1. For the same class, however, the complexity of private and nonprivate learning is $\Theta(d)$.

Our final point is that our lower bounds for $k$-ary distributions and general product distributions only require that the sampler generate a value *in the support* of the distribution with high probability. They thus apply to a weaker problem, similar in spirit to the interior point problem that forms the basis of lower bounds for private learning of thresholds [9].

Taken together, our results show that studying the complexity of generating a single sample helps us understand the true source of difficulty for certain tasks and sheds light on when we might be able to engage in nontrivial statistical tasks with very little data.

**Limitations of our results and open questions**   Our work raises many questions about the complexity of private sampling. First, our upper bounds achieve only the minimal goal of generating a single observation. In most settings, one would presumably want to generate several such samples. One can do so by repeating our algorithms several times on disjoint subsamples, but in general this is not the best way to proceed. Second, we study only three classes of distributions. It is likely that the picture of what is possible for many classes is more complex and nuanced. It would be interesting, for example, to study private sampling for Gaussian distributions, since they demonstrate intriguing data/accuracy tradeoffs for nonprivate sampling [4].

**Societal impact**  We study the feasibility of basic inference under privacy constraints. Our work is motivated by societal concerns, but focused on fundamental theoretical limits. We do not anticipate direct practical impact.

## 1.2  An overview of our proofs and techniques

For both algorithms and lower bounds, our results require the development of new techniques. On the algorithmic side, we take advantage of the fact that sampling algorithms need only be correct on average over samples drawn from a given distribution. One useful observation that underlies our positive results is that sampling based on an *unbiased* estimate $\hat{P}$ of a probability distribution $P$ (in the sense that $\mathbb{E}[\hat{P}(u)] = P(u)$ for all elements $u$ in the universe $\mathcal{U}$, where the expectation is taken over the randomness in the dataset and the coins of the algorithm) has zero error, even though the learning error, e.g., $d_{TV}(\hat{P}, P)$ might be large. For product distributions with bounded biases, we also exploit the randomness of the sampling process itself to gain privacy without explicitly adding noise.

For negative results, we cannot generally use existing lower bounds for learning or estimation, because of a fundamental obstacle. The basic framework used in proving most lower bounds on sample complexity of learning problems is based on *identifiability*: to show that a large sample is required to learn class $\mathcal{C}$, one first finds a set of distributions $P_1, ...., P_t$ in the class $\mathcal{C}$ that are far apart from each other and then shows that the output of any sufficiently accurate learning algorithm allows an outside observer to determine exactly which distribution in the collection generated the data. The final step is to show that algorithms in a given family (say, differentially private algorithms with a certain sample size) cannot reliably identify the correct distribution. This general approach is embodied in recipes such as Fano's, Assouad's, and Le Cam's methods from classical statistics (see, e.g., [3] for a summary of these methods and their differentially private analogues). For many sampling problems, the identifiability approach breaks down: a single observation is almost never enough to identify the originating distribution.

One of our approaches to proving lower bounds is to leverage ways in which the algorithm's output directly signals a failure to sample correctly. For instance, our lower bound for $k$-ary distribution relies on the fact that an $\alpha$-accurate sampler must produce a value in the support of the true distribution $P$ with high probability. Another approach is to reduce from other distribution (sampling or estimation) problems. For example, our lower bound for product distributions with bounded biases is obtained via a reduction from an estimation problem, by observing that a small number of samples from a nearby distribution suffices for a very weak estimate of the underlying distribution's attribute biases.

We break down our discussion of techniques according to the specific distribution classes we consider.

**Distributions on** $[k]$  For the class of distributions on $[k]$, Theorem 1.4 shows that $\alpha$-accurate $(\varepsilon, 0)$-differentially private sampling can be performed with a dataset of size $O(\frac{k}{\alpha\varepsilon})$. Our private sampler computes, for each $j \in [k]$, the proportion $\hat{P}_j$ of its dataset that is equal to $j$, adds Laplace noise to each count, uses $L_1$ projection to obtain a valid vector of probabilities $\tilde{P} = (\tilde{P}_1, \ldots, \tilde{P}_k)$, and finally outputs an element of $[k]$ sampled according to $\tilde{P}$.

Theorem 1.5 provides a matching lower bound on $n$ that holds for all $(\varepsilon, \delta)$-differentially private algorithms with $\delta = o(1/n)$. We prove our lower bound separately for Bernoulli distributions and for discrete distributions with support size $k \geq 3$, using different analyses. For Bernoulli distributions, we first exploit the group privacy for differentially private algorithms and the fact that the sampler must be accurate for the Bernoulli distribution $\mathrm{Ber}(0)$ to show that, on input with $t$ ones, a differentially private sampler outputs 1 with probability at most $2\alpha e^{\varepsilon t}$. Then we consider $P = \mathrm{Ber}(10\alpha)$. We use $\alpha$-accuracy in conjunction with group privacy to give a lower and an upper bound on the probability of the output being 1 when the input is drawn i.i.d. from $P$. This allows us to relate the parameters involved in order to obtain the desired lower bound on $n$.

The lower bound for distributions on $[k]$ with $k \geq 3$ is more involved. We start by identifying general properties of samplers that allow us to restrict our attention to relatively simple algorithms. First, we observe that every sampler can be converted to a *Poisson algorithm*, that is, an algorithm that, instead of receiving an input of a fixed size, gets an input with the number of records that follows a Poisson distribution. This observation allows us to use a standard technique called *Poissonization* that

makes the empirical frequencies of different elements independent. Next, we observe that privacy for samplers can be easily amplified, so that we can assume w.l.o.g. that $\varepsilon$ is small. Finally, we observe that every sampler for the class of $k$-ary distributions can be converted to a *frequency-count-based algorithm*. A sampler is *frequency-count-based* if the probability it outputs a specific element depends only on the number of occurrences of this element in its input and the frequency counts[2] of the input (that is, the number of elements that occur zero times, once, twice, thrice, and so on). Frequency-count-based algorithms have been studied for a long time in the context of understanding properties of distributions (see, e.g., [5, 6, 20]).

Equipped with the three observations, we restrict our attention to Poisson, frequency-count-based algorithms, with small $\varepsilon$ in the privacy guarantee. In contrast to our lower bound for Bernoulli samplers, we show that when the support size is at least 3 and the dataset size, $n$, is too small, the sampler is likely to output an element outside of the support of the input distribution $P$. Here, we exploit group privacy, which implies that the probability that a sampler outputs a specific element which appears $j$ times in its input differs by at most a factor of $e^{\varepsilon j}$ from the probability that it outputs a specific element that does not appear in the input. Then we consider a distribution $P$ that has most of its mass (specifically, $1 - O(\alpha)$) on a special element, and the remaining mass spread uniformly among half of the remaining domain elements. That is, $P$ is a mixture of a unit distribution on the special element and a uniform distribution on half of the remaining elements. We show that, when the dataset size is too small, the sampler is nearly equally likely to output any non-special element. But it has to output non-special elements with probability $\Omega(\alpha)$ to be $\alpha$-accurate. This means that when the database size is too small, the sampler outputs a non-special element outside the support of $P$ with probability $\Omega(\alpha)$. The details of the proof are quite technical and appear in Section 2.

**Product distributions**  Our private sampler for product distributions over $\{0, 1\}^d$, used to prove Theorem 1.6, builds on the recursive private preconditioning procedure designed by Kamath et al. [17] for learning this distribution class. In our case, the sampler gets a dataset of size which is asymptotically smaller than necessary for learning this distribution class in some important parameter regimes. Let $(p_1, \ldots, p_d)$ be the attribute biases for the product distribution $P$ from which the data is drawn. For simplicity, assume w.l.o.g. that all the marginal biases $p_j$ are less than 1/2. The main idea in [17] is that smaller biases have lower sensitivity in the following sense: if we know that a set of attributes has biases $p_j$ that are all at most some bound $u$ then, since the data is generated from a product distribution, the number of ones in those attributes should at most $2ud$ with high probability. We can enforce this bound on the number of ones in those coordinates by truncating each row appropriately, and thus learn the biases of those coordinates to higher accuracy than we knew before. Building on that idea, we partition the input into smaller datatsets and run our algorithm in rounds, each using fresh data, a different truncation threshold, and noise of different magnitude.

Our algorithm consists of two phases. In the bucketing phase, we use half of the dataset and the technique of [17] to get rough *multiplicative* estimates of all biases $p_j$ except the very small ones (where $p_j < 1/d$). This allows us to partition the coordinates into $\log(d)$ buckets, where biases within each bucket are similar. We show this crude estimation only requires $n = \tilde{O}(d/\alpha\varepsilon)$. In the sampling phase, we use the buckets to generate our output sample. For each bucket, we can get a fresh estimate of the biases using the other half of the dataset and, again, the technique from [17] to scale the noise proportionally to the upper bound on the biases that bucket. These estimates are essentially unbiased. Flipping $d$ coins independently according to the estimated biases produces an observation with essentially the correct distribution.

The proof of our lower bound for general product distributions proceeds via reduction from sampling of $k$-ary distributions for $k = d$. In contrast, lower bounds for private *learning* of product distributions rely on fingerprinting codes (building on the framework of Bun et al. [10]). Although fingerprinting codes are indeed useful when reasoning about samplers for distributions with bounded biases (discussed below), our approach relies, instead, on the fact that samplers must distinguish coordinates with bias 0 from coordinates with small bias. Specifically, given a distribution $P$ on $[2k]$ that is uniform on a subset of $[2k]$ of size $k$, we define a product distribution $P'$ on $\{0, 1\}^{2k}$ with biases $p_j = P(j) = \frac{1}{k}$ for $j \in S$ and $p_j = 0$ otherwise. We use Poissonization and coupling between Poisson and binomial distributions to design a reduction that, given observations from $P$, first creates an appropriately distributed sample of almost the same size drawn according to $P'$, then runs a

---

[2]The vector of frequency counts is called a *fingerprint* or a *histogram* in previous work.

hypothetical sampler for product distributions to get a vector in $\{0, 1\}^{2k}$ (drawn roughly according to $P'$), and finally converts that vector back to a single element of $[2k]$ distributed roughly as $P$. The details are subtle since most draws from $P'$ will not have exactly one nonzero element—this complicates conversion in both directions.

**Distributions with bounded bias**    Interestingly, our sampler for product distributions with bounded bias does not directly add noise to data. It performs the following step independently for each attribute: compute the empirical mean of the sample in that coordinate, obtain the clipped mean by rounding the empirical mean to the interval $[1/4, 3/4]$, and sample a bit according to the clipped mean. The key idea in the analysis of accuracy is that, conditioned on no rounding (that is, the empirical mean already being in the required interval), the new bit is sampled from the correct distribution, and rounding occurs with small probability. We argue that this sampler is $(4/n, 0)$-differentially private for the case when $d = 1$. For larger $d$, the sampler is a composition of $d$ differentially private algorithms, and the main bound follows from compositions theorems (and conversions between different variants of differential privacy).

The lower bound for this class proceeds by a reduction from the following weak estimation problem: Given samples from a product distribution with biases $p_1, p_2, ..., p_d$, output estimates $\tilde{p}_1, ..., \tilde{p}_d$ such that each $\tilde{p}_i$ is within additive error $\frac{1}{20}$ of $p_i$, that is, $|\tilde{p}_i - p_i| \leq \frac{1}{20}$, with probability at least $1 - \frac{1}{20}$ (where $\frac{1}{20}$ is just a sufficiently small constant). This problem is known to require datasets of size $\tilde{\Omega}(\sqrt{d})$ for differentially private algorithms [10]. However, nonprivate algorithms require only $O(1)$ data records to solve this same problem! We can thus reduce from estimation to sampling with very little overhead: Given a private sampler, we run it a constant number of times on disjoint datasets to obtain enough samples to (nonprivately) estimate each of the $p_i$'s. The nonprivate estimation is a postprocessing of an output of a differentially private algorithm, so the overall algorithm is differentially private. Some care is required in the reduction, since we must ensure that the nonprivate estimation algorithm is robust (i.e., works even when the samples are only close in TV distance to the correct distribution) and that the lower bound of [10] applies even when the biases $p_i$ lie in $[\frac{1}{3}, \frac{2}{3}]$.

## 1.3   Related work

To the best of our knowledge, the private sampling problem we formulate has not been studied previously. There is work on nonprivate sample-size amplification, in which an initial dataset of $n$ records is used to generate an output sample of size $n' > n$ from a nearby distribution [4]. Our formulation corresponds to the case where $n' = 1 \ll n$. There is also work on sampling (also called *simulation*) in a distributed setting [1, 2] where each of $n$ participants receives a single observation and is limited to $\ell \geq 1$ bits of communication. We are not aware of a direct technical connection with our work, even though the distributed setting is closely tied to that of *local* differential privacy.

In the literature on differential privacy, there is work on private algorithms for sampling parameters from Bayesian posterior distributions [12, 13, 24, 22], often driven by the intuition that the randomness inherent in sampling provides some level of privacy "for free". Our algorithms for product distributions with bounded biases leverage a similar idea. We are not aware of any analysis of the Bayesian approach which provides guarantees along the lines of our formulation (Def. 1.1).

The most substantially related work in the privacy literature is on upper and lower bounds for learning and estimation tasks, dating back to Warner [23] (see [16, 18] for partial surveys). For learning $k$-ary distributions, the directly most relevant works are those of [11, 3], though lower bounds for estimating histograms were known as folklore since the mid-2000s. For product distributions, tight upper and lower bounds for $(\varepsilon, \delta)$-differentially private estimation in TV distance appear in [17]; the upper bound was later shown to be achievable by an $(\varepsilon, 0)$-differentially private algorithm [8]. Our upper bound for general product distributions adapts the technique of [17]; it remains open whether our bound is achievable with $\delta = 0$.

Less directly related is the line of work on the generation of synthetic datasets that match the data distribution in a set of specified characteristics, such as linear queries (e.g., [7]; see [21] for a tutorial). The goal in those works is to generate enough synthetic data to allow an analyst given only the output to estimate the specified characteristics—a much more difficult task than sampling, and one for which a different set of lower bound techniques apply [21].

## 2 The lower bound for $k$-ary distributions

In this section, we prove the lower bound for sampling from discrete distributions with the universe of size at least 3. As discussed in Section 1.2, we prove that $n = \Omega(k/\alpha\varepsilon)$. The crux of the proof is the case where the sampler is frequency-count-based, Poisson, and is $(\varepsilon, \delta)$-DP for small $\varepsilon$. The transformation from general samplers to this restricted type of sampler is presented in the supplementary materials. This transformation together with the following Lemma 2.1 completes the proof of Theorem 1.5 for discrete distributions with the universe of size at least 3.

**Lemma 2.1.** *Fix $k, n \in \mathbb{N}, \alpha \in (0, 0.02], \varepsilon \in (0, 1/\ln(1/\alpha)],$ and $\delta \in [0, 0.1\alpha\varepsilon/k]$. Let $\mathcal{C}_{2k+1}$ denote the class of discrete distributions over the universe $[2k + 1]$. If sampler $\mathcal{A}$ is $(\varepsilon, \delta)$-DP, frequency-count-based, and $\alpha$-accurate on class $\mathcal{C}_{2k+1}$ with dataset size distributed as $\mathsf{Po}(n)$, then $n > \frac{1}{60} \cdot \frac{k}{\alpha\varepsilon}$.*

*Proof.* We consider the following distribution $P$ over the universe $\mathcal{U} = [2k + 1]$. Fix $\alpha^* = 60\alpha$ and a set $S^* \subset [2k]$ of size $k$. Distribution $P$ has mass $\alpha^*/k$ on each element in $S^*$ and mass $1 - \alpha^*$ on the *special* element $2k + 1$.

Consider a sampler $\mathcal{A}$ satisfying the conditions of Lemma 2.1. Let $Q_{\mathcal{A},P}$ denote the output distribution of $\mathcal{A}$ when the dataset size $N \sim \mathsf{Po}(n)$ and the dataset $\mathbf{X} \sim P^{\otimes N}$. Observe that

$$d_{TV}(Q_{\mathcal{A},P}, P) \geq \Pr_{\substack{N \sim \mathsf{Po}(n) \\ \mathbf{X} \sim P^{\otimes N}}}[\mathcal{A}(\mathbf{X}) \notin Supp(P)]. \tag{1}$$

We show that if $n \leq \frac{k}{60\alpha\varepsilon}$ and $\varepsilon$ and $\delta$ are in the specified range, the right-hand side of (1) is large. We start by deriving a lower bound on $\Pr[\mathcal{A}(\mathbf{x}) \notin Supp(P)]$ for a fixed dataset $\mathbf{x}$ of a fixed size $N$. Let $p_{j,F(\mathbf{x})}$ be the probability that $\mathcal{A}$ outputs a specific element in $[2k]$ that occurs $j$ times in $\mathbf{x}$, where $F(\mathbf{x})$ is the frequency-count of $\mathbf{x}$; these probabilities are well-defined because $\mathcal{A}$ is frequency-count-based. Let $F_0^*(\mathbf{x})$ denote the number of elements in $[2k]$ that occur 0 times in $\mathbf{x}$ (excluding the special element $2k + 1$ from this count). By definition, $F_0^*(\mathbf{x}) \leq 2k$. Consequently,

$$\Pr[\mathcal{A}(\mathbf{x}) \notin Supp(P)] = k \cdot p_{0,F(\mathbf{x})} \geq \frac{1}{2} \cdot F_0^*(\mathbf{x}) \cdot p_{0,F(\mathbf{x})}. \tag{2}$$

The next claim uses the fact that $\mathcal{A}$ is $(\varepsilon, \delta)$-DP to show that the probability $p_{j,F(\mathbf{x})}$ cannot be much larger than the probability that $\mathcal{A}$ outputs a specific element in $\mathcal{U}$ that does not appear in $\mathbf{x}$.

**Claim 2.2.** *For all $(\varepsilon, \delta)$-DP samplers, frequency counts $f \in \mathbb{Z}^*$, and elements $j \in \mathcal{U}$,*

$$p_{j,f} \leq e^{\varepsilon j}\left(p_{0,f} + \frac{\delta}{\varepsilon}\right). \tag{3}$$

*Proof.* Consider a frequency count $f$ and a dataset $\mathbf{x}$ with $F(\mathbf{x}) = f$. Note that (3) is true trivially for all $j$ such that $F_j(\mathbf{x}) = 0$ because, in that case, $p_{j,F(\mathbf{x})}$ is set to 0.

Fix any $j \in \mathcal{U}$ with $F_j(\mathbf{x}) > 0$. Let $a$ be any element in $\mathcal{U}$ that occurs $j$ times in the dataset $\mathbf{x}$. Let $b$ be any element in $\mathcal{U}$ that is not in the support of the distribution $P$. Let $\mathbf{x}|_{a \to b}$ denote the dataset obtained by replacing every instance of $a$ in the dataset $\mathbf{x}$ with element $b$. By group privacy [15],

$$\Pr[\mathcal{A}(\mathbf{x}) = a] \leq e^{j\varepsilon} \Pr[\mathcal{A}(\mathbf{x}|_{a \to b}) = a] + \delta \cdot \frac{e^{\varepsilon j} - 1}{e^\varepsilon - 1}. \tag{4}$$

Note that the dataset $\mathbf{x}|_{a \to b}$ does not contain element $a$, since we've replaced every instance of it with $b$. Importantly, $F(\mathbf{x}|_{a \to b}) = F(\mathbf{x})$ because $b$ is outside of the support of the distribution $P$ and hence does not occur in $\mathbf{x}$. Since $\mathcal{A}$ is frequency-count-based and $F(\mathbf{x}) = F(\mathbf{x}|_{a \to b})$, we get that $p_{0,F(\mathbf{x})} = p_{0,F(\mathbf{x}|_{a \to b})}$. Substituting this into (4) and using the fact that $e^\varepsilon - 1 \geq \varepsilon$ for all $\varepsilon$, we get

$$p_{j,F(\mathbf{x})} \leq e^{j\varepsilon} \cdot p_{0,F(\mathbf{x})} + \delta \cdot \frac{e^{\varepsilon j} - 1}{e^\varepsilon - 1} \leq e^{\varepsilon j}\left(p_{0,f} + \frac{\delta}{\varepsilon}\right).$$

This completes the proof of Claim 2.2. $\qquad\square$

For a dataset $\mathbf{x}$ and $i \in [2k + 1]$, let $N_i(\mathbf{x})$ denote the number of occurrences of element $i$ in $\mathbf{x}$. Next, we give a lower bound on $\Pr[\mathcal{A}(\mathbf{x}) \notin Supp(P)]$ in terms of the counts $N_i(\mathbf{x})$.

**Claim 2.3.** *Let $N \in \mathbb{N}$ and $\mathbf{x} \in [2k+1]^N$ be a fixed dataset. Set $Y = \sum_{i \in S^*} \left[ e^{N_i(\mathbf{x})\varepsilon} \right]$. Then*

$$\Pr[\mathcal{A}(\mathbf{x}) \notin Supp(P)] \geq \frac{1}{2} \cdot \frac{\Pr[\mathcal{A}(\mathbf{x}) \in [2k]]}{1 + Y/k} - \frac{k\delta}{\varepsilon}.$$

*Proof.* In the following derivation, we use the fact that that an element $j \in [2k]$ that appears $j$ times in $\mathbf{x}$ is returned by $\mathcal{A}$ with probability $p_{j,F(\mathbf{x})}$, then split the elements into those that do not appear in $\mathbf{x}$ and those that do, next use the fact that all elements from $[2k]$ that appear in $\mathbf{x}$ must be in $S^*$, then apply Claim 2.2, and finally substitute $Y$ for $\sum_{i \in S^*} \left[ e^{N_i(\mathbf{x})\varepsilon} \right]$:

$$\Pr[\mathcal{A}(\mathbf{x}) \in [2k]] = \sum_{i \in [2k]} p_{N_i(\mathbf{x}),F(\mathbf{x})} = F_0^*(\mathbf{x}) \cdot p_{0,F(\mathbf{x})} + \sum_{i \in [2k] \cap \mathbf{x}} p_{N_i(\mathbf{x}),F(\mathbf{x})}$$

$$\leq F_0^*(\mathbf{x}) \cdot p_{0,F(\mathbf{x})} + \sum_{i \in S^*} p_{N_i(\mathbf{x}),F(\mathbf{x})}$$

$$\leq F_0^*(\mathbf{x}) \cdot p_{0,F(\mathbf{x})} + \sum_{i \in S^*} p_{0,F(\mathbf{x})} \cdot \left( e^{\varepsilon N_i(\mathbf{x})} + \frac{\delta}{\varepsilon} \right) \leq \left( F_0^*(\mathbf{x}) + Y \right) \left( p_{0,F(\mathbf{x})} + \frac{\delta}{\varepsilon} \right).$$

We rearrange the terms to get $p_{0,F(\mathbf{x})} \geq \dfrac{\Pr[\mathcal{A}(\mathbf{x}) \in [2k]]}{F_0^*(\mathbf{x}) + Y} - \dfrac{\delta}{\varepsilon}$.

Substituting this bound on $p_{0,F(\mathbf{x})}$ into (2), we obtain that $\Pr[\mathcal{A}(\mathbf{x}) \notin Support(P)]$ is at least

$$\frac{1}{2} \cdot \frac{F_0^*(\mathbf{x}) \Pr[\mathcal{A}(\mathbf{x}) \in [2k]]}{F_0^*(\mathbf{x}) + Y} - \frac{1}{2} \cdot \frac{F_0^*(\mathbf{x}) \cdot \delta}{\varepsilon} \geq \frac{1}{2} \cdot \frac{\Pr[\mathcal{A}(\mathbf{x}) \in [2k]]}{1 + Y/k} - \frac{k\delta}{\varepsilon},$$

where in the inequality, we used that $k \leq F_0^*(\mathbf{x}) \leq 2k$. This holds since the support of $P$ excludes $k$ elements from $[2k]$ and since $F_0^*(\mathbf{x})$ counts only elements from $[2k]$ that do not appear in $\mathbf{x}$. $\qquad\square$

Finally, we give a lower bound on the right-hand side of (1). Assume for the sake of contradiction that $n \leq \frac{k}{\alpha^* \varepsilon}$. By Claim 2.3,

$$\Pr_{N,\mathbf{X}}[\mathcal{A}(\mathbf{X}) \notin Supp(P)] \geq \mathbb{E}_{N,\mathbf{X}} \left[ \frac{1}{2} \cdot \frac{\Pr[\mathcal{A}(\mathbf{X}) \in [2k]]}{1 + Y/k} - \frac{k\delta}{\varepsilon} \right]$$

$$= \frac{1}{2} \cdot \mathbb{E}_{N,\mathbf{X}} \left[ \frac{\Pr[\mathcal{A}(\mathbf{X}) \in [2k]]}{1 + Y/k} \right] - \frac{k\delta}{\varepsilon}. \tag{5}$$

Next, we analyze the expectation in (5). Let $E$ be the event that $\frac{Y}{k} \leq e^3$. By the law of total expectation,

$$\mathbb{E}_{N,\mathbf{X}} \left[ \frac{\Pr[\mathcal{A}(\mathbf{X}) \in [2k]]}{1 + Y/k} \right] \geq \mathbb{E}_{N,\mathbf{X}} \left[ \frac{\Pr[\mathcal{A}(\mathbf{X}) \in [2k]]}{1 + Y/k} \Big| E \right] \Pr(E). \tag{6}$$

In Claims 2.4 and 2.5, we argue that both $\Pr(E)$ and $\mathbb{E}_{N,\mathbf{X}} \left[ \frac{\Pr[\mathcal{A}(\mathbf{X}) \in [2k]]}{1 + Y/k} \Big| E \right]$ are large.

**Claim 2.4.** *Suppose $n \leq \frac{k}{60\alpha\varepsilon}$. Let $E$ be the event that $\frac{Y}{k} \leq e^3$. Then $\Pr(E) \geq 1 - \alpha$.*

*Proof.* Recall that $Y$ was defined as $\sum_{i \in S^*} \left[ e^{N_i(\mathbf{x})\varepsilon} \right]$ for a fixed dataset $\mathbf{x}$. Now consider the case when dataset $\mathbf{X}$ is a random variable. If $N \sim \mathsf{Po}(n)$ and $\mathbf{X} \sim P^{\otimes N}$ then $N_i(\mathbf{X}) \sim \mathsf{Po}(\frac{\alpha^* n}{k})$ for all $i \in S^*$ and, additionally, the random variables $N_i(\mathbf{X})$ are mutually independent. When $\mathbf{X}$ is clear from the context, we write $N_i$ instead of $N_i(\mathbf{X})$. Now we calculate the moments of $\frac{Y}{k}$. For all $\lambda > 0$,

$$\mathbb{E}_{\substack{N \sim \mathsf{Po}(n) \\ \mathbf{X} \sim P^{\otimes N}}} \left[ \left( \frac{Y}{k} \right)^\lambda \right] = \mathbb{E}_{\substack{N \sim \mathsf{Po}(n) \\ \mathbf{X} \sim P^{\otimes N}}} \left[ \left( \frac{1}{k} \sum_{i \in S^*} e^{N_i(\mathbf{X})\varepsilon} \right)^\lambda \right] = \mathbb{E}_{N_1,\cdots,N_k \sim \mathsf{Po}(\frac{\alpha^* n}{k})} \left[ \left( \frac{1}{k} \sum_{i \in S^*} e^{N_i \varepsilon} \right)^\lambda \right]. \tag{7}$$

Finally, we bound the probability of event $E$. Set $c = e^3$ and $\lambda = \ln \frac{1}{\alpha}$. By definition of $E$,

$$\Pr(\overline{E}) = \Pr\left(\frac{Y}{k} \geq c\right) = \Pr\left(\left(\frac{Y}{k}\right)^\lambda \geq c^\lambda\right) \leq \frac{1}{c^\lambda} \cdot \mathop{\mathbb{E}}_{\substack{N \sim \mathsf{Po}(n) \\ \mathbf{X} \sim P^{\otimes N}}} \left[\left(\frac{Y}{k}\right)^\lambda\right]$$

$$\leq \frac{1}{c^\lambda} \cdot \mathop{\mathbb{E}}_{N_1, \cdots, N_k \sim \mathsf{Po}(\frac{\alpha^* n}{k})} \left[\left(\frac{1}{k}\sum_{i \in S^*} e^{N_i \varepsilon}\right)^\lambda\right] \leq \frac{1}{c^\lambda} \cdot \mathop{\mathbb{E}}_{N_1 \sim \mathsf{Po}(\frac{\alpha^* n}{k})} \left[\left(e^{N_1 \varepsilon}\right)^\lambda\right] \quad (8)$$

$$= c^{-\lambda} \cdot e^{\frac{\alpha^* n}{k}(e^{\lambda \varepsilon}-1)} \leq e^{-3\lambda} \cdot e^{\frac{(e^{\lambda\varepsilon}-1)}{\varepsilon}} \leq e^{-3\lambda} \cdot e^{2\lambda} = e^{-\lambda} = e^{-\ln(1/\alpha)} = \alpha, \quad (9)$$

where we use $\lambda > 0$ in the second equality, then apply Markov's inequality. To get the inequalities in (8), we apply (7) and then use the fact that the moments of the average of random variables is less than the moment of a single random variable (the proof of this fact is in the supplementary material). To get (9), we use the moment generating function of a Poisson random variable, and then we substitute $c = e^3$ and use the assumption that $n \leq \frac{k}{60\alpha\varepsilon} = \frac{k}{\alpha^*\varepsilon}$. The second inequality in (9) holds because $\lambda = \ln \frac{1}{\alpha}$ and $\varepsilon \in (0, 1/\ln\frac{1}{\alpha}]$, so $\lambda\varepsilon \leq 1$ and hence $e^{\lambda\varepsilon} \leq 1 + 2\lambda\varepsilon$. The final expression is obtained by substituting the value of $\lambda$. We get that $\Pr(E) \geq 1 - \alpha$, completing the proof of Claim 2.4. $\quad\square$

**Claim 2.5.** $\mathop{\mathbb{E}}_{\substack{N \sim \mathsf{Po}(n) \\ \mathbf{X} \sim P^{\otimes N}}} \left[\frac{\Pr[\mathcal{A}(\mathbf{X}) \in [2k]]}{1 + Y/k} \Big| E\right] \geq 2.3\alpha.$

*Proof.* When event $E$ occurs, $1 + \frac{Y}{k} \leq 1 + e^3 < 22$. Then

$$\mathop{\mathbb{E}}_{N,\mathbf{X}} \left[\frac{\Pr[\mathcal{A}(\mathbf{X}) \in [2k]]}{1 + Y/k} \Big| E\right] > \mathop{\mathbb{E}}_{N,\mathbf{X}} \left[\frac{\Pr[\mathcal{A}(\mathbf{X}) \in [2k]]}{22} \Big| E\right] = \frac{1}{22} \cdot \mathop{\mathbb{E}}_{N,\mathbf{X}} \left[\Pr[\mathcal{A}(\mathbf{X}) \in [2k]] \mid E\right]. \tag{10}$$

By the product rule,

$$\Pr[\mathcal{A}(\mathbf{X}) \in [2k]] \mid E] = \frac{\Pr[\mathcal{A}(\mathbf{X}) \in [2k] \wedge E]}{\Pr[E]} \geq \Pr[\mathcal{A}(\mathbf{X}) \in [2k] \wedge E] \geq \Pr[\mathcal{A}(\mathbf{X}) \in [2k]] - \Pr[\overline{E}].$$

Substituting this into (10) and recalling that $\alpha^* = 60\alpha$, we get

$$\mathop{\mathbb{E}}_{N,\mathbf{X}} \left[\frac{\Pr[\mathcal{A}(\mathbf{X}) \in [2k]]}{1 + Y/k} \Big| E\right] \geq \frac{1}{22} \cdot \mathop{\mathbb{E}}_{N,\mathbf{X}} \left[\Pr[\mathcal{A}(\mathbf{X}) \in [2k]] - \Pr[\overline{E}]\right] \geq \frac{1}{22} \cdot \left(\alpha^* - \alpha - \alpha\right) \geq 2.3\alpha,$$

since sampler $\mathcal{A}$ is $\alpha$-accurate on $P$, and $P$ has mass $\alpha^*$ on $[2k]$, and by Claim 2.4. $\quad\square$

Combining (1), (5), and (6), applying Claims 2.4 and 2.5, and recalling that $\delta \leq 0.1 \cdot \alpha\varepsilon/k$, we get

$$d_{TV}(P, Q_{\mathcal{A},P}) \geq \Pr_{N,\mathbf{X}}[\mathcal{A}(\mathbf{X}) \notin Supp(P)] \geq \frac{1}{2} \cdot \mathop{\mathbb{E}}_{N,\mathbf{X}} \left[\frac{\Pr[\mathcal{A}(\mathbf{X}) \in [2k]]}{1 + Y/k}\right] - \frac{k\delta}{\varepsilon}$$

$$\geq \frac{1}{2} \cdot \mathop{\mathbb{E}}_{N,\mathbf{X}} \left[\frac{\Pr[\mathcal{A}(\mathbf{X}) \in [2k]]}{1 + Y/k} \Big| E\right] \Pr(E) - 0.1\alpha \geq \frac{1}{2} \cdot 2.3\alpha \cdot \left(1 - \alpha\right) - 0.1\alpha > \alpha,$$

where the last inequality holds since $\alpha \leq 0.02$. This contradicts $\alpha$-accuracy of $\mathcal{A}$ on datasets of size $\mathsf{Po}(n)$, where $n \leq \frac{k}{\alpha^*\varepsilon}$, and completes the proof of Lemma 2.1. $\quad\square$

## Acknowledgments and Disclosure of Funding

We are grateful for helpful conversations with Clément Canonne, Thomas Steinke, and Jonathan Ullman. Sofya Raskhodnikova was partially supported by NSF award CCF-1909612. Satchit Sivakumar was supported in part by NSF award CNS-2046425, as well as Cooperative Agreement CB20ADR0160001 with the Census Bureau. Adam Smith and Marika Swanberg were supported in part by NSF award CCF-1763786 as well as a Sloan Foundation research award. The views expressed in this paper are those of the authors and not those of the U.S. Census Bureau or any other sponsor.

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
