# Supplementary Materials

---

## Differentially Private Sampling from Distributions

These supplementary materials are organized as follows. Section A collects standard definitions and mathematical tools. Next, in Section B, we describe general transformations of samplers that we use in our lower bounds. In Section C.3, we prove upper and lower bounds for the task of private sampling from $k$-ary distributions, corresponding to Theorems 1.4 and 1.5 in the introduction. In Section D, we prove upper bounds for private sampling from product distributions over $\{0,1\}^d$, corresponding to Theorem 1.6. We defer the proof of Theorem 1.7 to the full version of the paper. In Section E, we present our upper and lower bounds for private sampling from product distributions with bounded attribute biases, corresponding to Theorems 1.8 and 1.9 in the introduction. Finally, in Section F, we discuss some standard results that we use in our proofs and, in Section G, we state some results from other papers that we use in our proofs.

## A  Definitions

### A.1  Differential Privacy

A dataset $\mathbf{x} = (x_1, \ldots, x_n) \in \mathcal{U}^n$ is a vector of elements from universe $\mathcal{U}$. Two datasets are *neighbors* if they differ in at most one coordinate. Informally, differential privacy requires that an algorithm's output distributions are similar on all pairs of neighboring datasets. We use two different variants of differential privacy. The first one (and the main one used in this paper) is the standard definition of differential privacy.

**Definition A.1** (Differential Privacy [17, 16]). *A randomized algorithm* $\mathcal{A} : \mathcal{U}^n \to \mathcal{Y}$ *is* $(\varepsilon, \delta)$-*differentially private if for every pair of neighboring datasets* $\mathbf{x}, \mathbf{x}' \in \mathcal{U}^n$ *and for all subsets* $Y \subseteq \mathcal{Y}$,

$$\Pr[\mathcal{A}(\mathbf{x}) \in Y] \le e^\varepsilon \cdot \Pr[\mathcal{A}(\mathbf{x}') \in Y] + \delta.$$

In addition to standard differential privacy (Definition 1.3), we use a variant called *zero-mean concentrated differential privacy* [10] that is defined in terms of Rényi divergence.

**Definition A.2** (Rényi divergence). *Consider two probability distributions* $P$ *and* $Q$ *over a discrete domain* $S$. *Given a positive* $\alpha \ne 1$, *Rényi divergence of order* $\alpha$ *of distributions* $P$ *and* $Q$ *is*

$$D_\alpha(P||Q) = \frac{1}{1-\alpha} \log \left( \sum_{x \in S} P(x)^\alpha Q(x)^{1-\alpha} \right).$$

**Definition A.3** (Zero-Mean Concentrated Differential Privacy (zCDP) [10]). *A randomized algorithm* $\mathcal{A} : \mathcal{U}^n \to \mathcal{Y}$ *is* $\rho$-*zCDP if for every pair of neighboring datasets* $\mathbf{x}, \mathbf{x}' \in \mathcal{U}^n$,

$$\forall \alpha \in (1, \infty) \quad D_\alpha \left( \mathcal{A}(\mathbf{x}) || \mathcal{A}(\mathbf{x}') \right) \le \rho\alpha,$$

*where* $D_\alpha(\mathcal{A}(\mathbf{x})||\mathcal{A}(\mathbf{x}'))$ *is the* $\alpha$-*Rényi divergence between* $\mathcal{A}(\mathbf{x})$ *and* $\mathcal{A}(\mathbf{x}')$.

**Lemma A.4** (Relationships Between $(\varepsilon, \delta)$-Differential Privacy and $\rho$-CDP [10]). *For every* $\varepsilon \ge 0$,

1. *If* $\mathcal{A}$ *is* $(\varepsilon, 0)$-*differentially private, then* $\mathcal{A}$ *is* $\frac{\varepsilon^2}{2}$-*zCDP.*

2. *If* $\mathcal{A}$ *is* $\frac{\varepsilon^2}{2}$-*zCDP, then* $\mathcal{A}$ *is* $\left( \frac{\varepsilon^2}{2} + \varepsilon\sqrt{2\log(1/\delta)}, \delta \right)$-*differentially private for every* $\delta > 0$.

Both definitions of differential privacy are closed under post-processing.

**Lemma A.5** (Post-Processing [17, 10]). *If $\mathcal{A} : \mathcal{U}^n \to \mathcal{Y}$ is $(\varepsilon, \delta)$-differentially private, and $\mathcal{B} : \mathcal{Y} \to \mathcal{Z}$ is any randomized function, then the algorithm $\mathcal{B} \circ \mathcal{A}$ is $(\varepsilon, \delta)$-differentially private. Similarly, if $\mathcal{A}$ is $\rho$-zCDP then the algorithm $\mathcal{B} \circ \mathcal{A}$ is $\rho$-zCDP.*

Importantly, both notions of differential privacy are closed under adaptive composition. For a fixed dataset $\mathbf{x}$, *adaptive composition* states that the results of a sequence of computations satisfies differential privacy even when the chosen computation $\mathcal{A}_t(\cdot)$ at time $t$ depends on the outcomes of previous computations $\mathcal{A}_1(\mathbf{x}), \ldots, \mathcal{A}_{t-1}(\mathbf{x})$. Under adaptive composition, the privacy parameters add up.

**Definition A.6** (Composition of $(\varepsilon, \delta)$-differential privacy and $\rho$-zCDP [17, 10]). *Suppose $\mathcal{A}$ is an adaptive composition of differentially private algorithms $\mathcal{A}_1, \ldots, \mathcal{A}_T$.*

1. *If for each $t \in [T]$, algorithm $\mathcal{A}_t$ is $(\varepsilon_t, \delta_t)$-differentially private, then $\mathcal{A}$ is $(\sum_t \varepsilon_t, \sum_t \delta_t)$-differentially private.*

2. *If for each $t \in [T]$, algorithm $\mathcal{A}_t$ is $\rho_t$-zCDP, then $\mathcal{A}$ is $(\sum_t \rho_t)$-zCDP.*

Standard $(\varepsilon, \delta)$-differential privacy protects the privacy of groups of individuals.

**Lemma A.7** (Group Privacy [17]). *Every $(\varepsilon, \delta)$-differentially private algorithm $\mathcal{A}$ is $\left(k\varepsilon, \delta \frac{e^{k\varepsilon}-1}{e^{\varepsilon}-1}\right)$-differentially private for groups of size $k$. That is, for all datasets $\mathbf{x}, \mathbf{x}'$ such that $\|\mathbf{x} - \mathbf{x}'\|_0 \leq k$ and all subsets $Y \subseteq \mathcal{Y}$,*

$$\Pr[\mathcal{A}(\mathbf{x}) \in Y] \leq e^{k\varepsilon} \cdot \Pr[\mathcal{A}(\mathbf{x}') \in Y] + \delta \cdot \frac{e^{k\varepsilon}-1}{e^{\varepsilon}-1}.$$

**Laplace Mechanism** Our algorithms use the standard Laplace Mechanism to ensure differential privacy.

**Definition A.8** (Laplace Distribution). *The Laplace distribution with parameter $b$ and mean 0, denoted by $\mathsf{Lap}(b)$, is defined for all $x \in \mathbb{R}$ and has probability density*

$$h(x) = \frac{1}{2b} e^{-\frac{|x|}{b}}.$$

**Definition A.9** ($\ell_1$-Sensitivity). *Let $f : \mathcal{U}^n \to \mathbb{R}^d$ be a function. Its $\ell_1$-sensitivity is*

$$\Delta_f = \max_{\substack{\mathbf{x}, \mathbf{x}' \in \mathcal{U}^n \\ \mathbf{x}, \mathbf{x}' \, neighbors}} \|f(\mathbf{x}) - f(\mathbf{x}')\|_1.$$

**Lemma A.10** (Laplace Mechanism). *Let $f : \mathcal{U}^n \to \mathbb{R}^d$ be a function with $\ell_1$-sensitivity $\Delta_f$. Then the Laplace mechanism is algorithm*

$$\mathcal{A}_f(\mathbf{x}) = f(\mathbf{x}) + (Z_1, \ldots, Z_d),$$

*where $Z_i \sim \mathsf{Lap}\left(\frac{\Delta_f}{\varepsilon}\right)$. Algorithm $\mathcal{A}_f$ is $(\varepsilon, 0)$-differentially private.*

**Gaussian Mechanism** Our algorithms also use the common Gaussian Mechanism to ensure differential privacy.

**Definition A.11** (Gaussian Distribution). *The Gaussian distribution with parameter $\sigma$ and mean 0, denoted $\mathcal{N}(\sigma)$, is defined for all $x \in \mathbb{R}$ and has probability density*

$$h(x) = \frac{1}{\sigma\sqrt{2\pi}} e^{-\frac{x^2}{2\sigma^2}}.$$

**Definition A.12** ($\ell_2$-Sensitivity). *Let $f : \mathcal{U}^n \to \mathbb{R}^d$ be a function. Its $\ell_2$-sensitivity is*

$$\Delta_f = \max_{\substack{\mathbf{x}, \mathbf{x}' \in \mathcal{U} \\ \mathbf{x}, \mathbf{x}' \, neighbors}} \|f(\mathbf{x}) - f(\mathbf{x}')\|_2.$$

**Lemma A.13** (Gaussian Mechanism)**.** *Let $f : \mathcal{U}^n \to \mathbb{R}^d$ be a function with $\ell_2$-sensitivity $\Delta_f$. Then the Gaussian mechanism is algorithm*

$$\mathcal{A}_f(\mathbf{x}) = f(\mathbf{x}) + (Z_1, \ldots, Z_d),$$

*where $Z_i \sim \mathcal{N}\left(\left(\frac{\Delta_f}{\sqrt{2\rho}}\right)^2 \cdot \mathbb{I}\right)$. Algorithm $\mathcal{A}_f$ is $\rho$-zCDP.*

## A.2   Distributions

Additionally, we use the Bernoulli and Poisson distributions and total variation distance.

**Definition A.14** (Bernoulli Distribution)**.** *The Bernoulli distribution with bias $p \in [0, 1]$, denoted $\mathsf{Ber}(p)$, is defined for $x \in \{0, 1\}$ with density*

$$h(x) = \begin{cases} p & \text{if } x = 1; \\ 1 - p & \text{if } x = 0. \end{cases}$$

**Definition A.15** (Binomial Distribution)**.** *The Binomial distribution with parameter $n, p$, denoted $Bin(n, p)$, is defined for all non-negative integers $x$ such that $x \leq n$ with density*

$$h(x) = \binom{n}{x} p^x (1 - p)^{n-x}$$

**Definition A.16** (Poisson Distribution)**.** *The Poisson distribution with parameter $\lambda$, denoted $\mathsf{Po}(\lambda)$, is defined for all $x \in \mathbb{R}$ with probability density*

$$h(x) = \frac{\lambda^x e^{-x}}{x!}.$$

**Definition A.17** (Total Variation Distance)**.** *Let $P$ and $Q$ be discrete probability distributions over some domain $S$. Then*

$$d_{TV}(P, Q) := \frac{1}{2}\|P - Q\|_1 = \sup_{E \subseteq S} |\Pr_P(E) - \Pr_Q(E)|.$$

We use the fact that the total variation distance between two product distributions is subadditive.

**Definition A.18** (Subadditivity of TV Distance for Product Distributions)**.** *Let $P$ and $Q$ be product distributions over some domain $S$. Let $P^1, \ldots, P^d$ be the marginal distributions of $P$ and $Q^1, \ldots, Q^d$ be the marginal distributions over $Q$. Then,*

$$d_{TV}(P, Q) \leq \sum_{i=1}^d d_{TV}(P^i, Q^i)$$

**Definition A.19** (KL Divergence)**.** *Let $P$ and $Q$ be discrete probability distributions over some domain $S$. Then*

$$d_{KL}(P, Q) := \frac{1}{2} \sum_{x \in S} P(x) \log\left(\frac{P(x)}{Q(x)}\right)$$

**Claim A.20.** *For a Bernoulli distribution $\mathsf{Ber}(p)$, we can simplify the definition of $\alpha$-accuracy (Definition 1.1) of a sampler $\mathcal{A}$ with inputs of size $n$ to require that*

$$d_{TV}(Q_{\mathcal{A}, \mathsf{Ber}(p)}, \mathsf{Ber}(p)) = \left| \Pr_{\mathbf{X} \sim (\mathsf{Ber}(p))^{\otimes n}} [\mathcal{A}(\mathbf{X}) = 1] - p \right| = \left| \mathbb{E}_{\mathbf{X} \sim (\mathsf{Ber}(p))^{\otimes n}} [\mathbb{1}_{\mathcal{A}(\mathbf{X})=1}] - p \right| \leq \alpha.$$

# B    Properties of Samplers

In this section, we describe three general transformations of samplers that allow us to assume without loss of generality that samplers can take a certain specific form. All three transformations are used in our lower bound proofs.

## B.1    General Samplers to Poisson Samplers

In the first transformation, we show that any private sampling task can be performed by an algorithm that gets a dataset with size distributed as a Poisson random variable instead of getting a dataset of a fixed size. This enables the use of the technique called *Poissonization* to break dependencies between quantities that arise in trying to reason about samplers. Recall that $\mathsf{Po}(\lambda)$ denotes a Poisson distribution with mean $\lambda$.

**Lemma B.1.** *If there exists an $(\varepsilon, \delta)$-differentially private sampler $\mathcal{A}$ that is $\alpha$-accurate on distribution class $\mathcal{C}$ for datasets of size $n$, then there exists an $(\varepsilon, \delta)$-differentially private sampler $\mathcal{A}_{\mathsf{Po}}$ that is $(\alpha + e^{-n/6})$-accurate on class $\mathcal{C}$ for datasets of size distributed as $\mathsf{Po}(2n)$.*

*Proof.* Algorithm 1 is the desired sampler $\mathcal{A}_{\mathsf{Po}}$. It is $(\varepsilon, \delta)$-differentially private since $\mathcal{A}$ is $(\varepsilon, \delta)$-differentially private. Let $\mathbf{X}$ represent the random variable corresponding to the dataset fed to $\mathcal{A}$.

---
**Algorithm 1** Sampler $\mathcal{A}_{\mathsf{Po}}$ with dataset size $N \sim \mathsf{Po}(2n)$

> **Input:** dataset $\mathbf{x} = (x_1, \dots, x_N)$, universe $\mathcal{U}$, oracle access to $(\varepsilon, \delta)$-DP sampler $\mathcal{A}$, parameter $n$
> **Output:** $i \in \mathcal{U}$
1: Fix an element $s \in \mathcal{U}$.
2: **if** $N < n$ **then** $i = s$.
3: **else** $i \leftarrow \mathcal{A}(\mathbf{x})$
4: **return** $i$.

---

We use the following tail bound for Poisson random variables [12].

**Claim B.2** ([12])**.** *If $Y \sim \mathsf{Po}(\lambda)$, then $\Pr(Y \leq \lambda - y) \leq e^{-\frac{y^2}{2(\lambda + y)}}$ for all $y > 0$.*

Let event $E$ correspond to $N < n$. Let $\overline{E}$ represent the complement of $E$. Then

$$\Pr(E) \leq e^{-\frac{n}{6}} \tag{1}$$

by an application of Claim B.2. Let $P \in \mathcal{C}$ and $\mathbf{X} \sim P^{\otimes N}$. Then

$$
\begin{aligned}
d_{TV}(Q_{\mathcal{A}_{\mathsf{Po}}, P}, P) &= \frac{1}{2} \sum_{i \in \mathcal{U}} | \Pr_{N, \mathcal{A}_{\mathsf{Po}}, \mathbf{X}}(\mathcal{A}_{\mathsf{Po}}(\mathbf{X}) = i) - P(i)| \\
&= \frac{1}{2} \sum_{i \in \mathcal{U}} \left| \Pr_{N, \mathcal{A}_{\mathsf{Po}}, \mathbf{X}}(\mathcal{A}_{\mathsf{Po}}(\mathbf{X}) = i \wedge \overline{E}) + \Pr_{N, \mathcal{A}_{\mathsf{Po}}, \mathbf{X}}(\mathcal{A}_{\mathsf{Po}}(\mathbf{X}) = i \wedge E) - P(i) \left( \Pr_N(\overline{E}) + \Pr_N(E) \right) \right| \\
&\leq \frac{1}{2} \sum_{i \in \mathcal{U}} \left( \left| \Pr_{N, \mathcal{A}_{\mathsf{Po}}, \mathbf{X}}(\mathcal{A}_{\mathsf{Po}}(\mathbf{X}) = i \wedge \overline{E}) - P(i) \Pr_N(\overline{E}) \right| + \Pr_{N, \mathcal{A}_{\mathsf{Po}}, \mathbf{X}}(\mathcal{A}_{\mathsf{Po}}(\mathbf{X}) = i \wedge E) + \Pr_N(E) P(i) \right) \\
&= \frac{1}{2} \sum_{i \in \mathcal{U}} \left| \Pr_{N, \mathcal{A}_{\mathsf{Po}}, \mathbf{X}}(\mathcal{A}_{\mathsf{Po}}(\mathbf{X}) = i \mid \overline{E}) \Pr(\overline{E}) - P(i) \Pr_N(\overline{E}) \right| + \frac{1}{2} \cdot \left( \Pr_N(E) + \Pr_N(E) \right) \\
&= \frac{1}{2} \sum_{i \in \mathcal{U}} \Pr_N(\overline{E}) \cdot \left| \Pr_{\mathcal{A}, \mathbf{X}, N | \overline{E}}(\mathcal{A}(\mathbf{X}) = i \mid \overline{E}) - P(i) \right| + \Pr_N(E) \\
&\leq \frac{1}{2} \sum_{i \in \mathcal{U}} \left| \Pr_{\mathcal{A}, \mathbf{X}}(\mathcal{A}(\mathbf{X}) = i \mid \overline{E}) - P(i) \right| + \Pr_N(E) \\
&\leq \alpha + e^{-n/6},
\end{aligned}
$$

where the first equality is by the definition of total variation distance, the second equality is because $\Pr(\overline{E}) + \Pr(E) = 1$ and because $\Pr(a) = \Pr(a, E) + \Pr(a, \overline{E})$ for every event $a$, the first inequality is because of the triangle inequality, the third equality is by the product rule and by marginalizing over the outputs, the fourth inequality is because when $N > n$, $\mathcal{A}_{\mathsf{Po}}$ sets the output to be $\mathcal{A}(x)$, the second inequality is by the fact that $\Pr_N(\overline{E}) \leq 1$, and the final inequality is by (1) and the fact that the sampler $\mathcal{A}$ is $\alpha$-accurate when it gets any fixed number of samples larger than $n$. Hence, $\mathcal{A}_{\mathsf{Po}}$ is $(\alpha + e^{-n/6})$-accurate on $\mathcal{C}$. $\qquad\square$

## B.2 Privacy Amplification for Samplers

Our second general transformation shows how to amplify privacy (that is, decrease privacy parameters) of a sampler by subsampling its input. The transformation does not affect the accuracy. The following lemma quantifies how the privacy parameters and the dataset size are affected by privacy amplification. This result is needed in the proof of the lower bound for $k$-ary distributions, because the main technical lemma in that proof (Lemma C.3) only applies to samplers with small $\varepsilon$. It is well known that subsampling amplifies differential privacy (see, e.g., [23, 22]).

**Lemma B.3.** *Fix $\varepsilon \in (0, 1], \delta \in (0, 1)$, and $\beta \in (0, 1)$. If there exists an $(\varepsilon, \delta)$-differentially private sampler $\mathcal{A}$ that is $\alpha$-accurate on distribution class $\mathcal{C}$ for datasets of size distributed as $\mathsf{Po}(n)$ then there exists an $(\varepsilon\beta, \delta\frac{\beta}{2})$-differentially private sampler $\mathcal{A}_{\beta/2}$ that is $\alpha$-accurate on class $\mathcal{C}$ for datasets of size distributed as $\mathsf{Po}\left(n \cdot \frac{2}{\beta}\right)$.*

*Proof.* We construct $\mathcal{A}_{\beta/2}$ from sampler $\mathcal{A}$ as follows: Given a dataset $\mathbf{x}$, sampler $\mathcal{A}_{\beta/2}$ subsamples each record in $\mathbf{x}$ independently with probability $\beta/2$ to get a new dataset $\mathbf{x}^*$ and then returns $\mathcal{A}(\mathbf{x}^*)$.

First, we argue that if sampler $\mathcal{A}$ is $(\varepsilon, \delta)$-differentially private, then sampler $\mathcal{A}_{\beta/2}$ is $(\varepsilon\beta, \delta\frac{\beta}{2})$-differentially private. This follows from [22, Theorem 1] which we state as Theorem G.2 in the appendix. By Theorem G.2, algorithm $\mathcal{A}_{\beta/2}$ is $(\varepsilon', \delta \cdot \frac{\beta}{2})$-differentially private with

$$\varepsilon' = \ln\left(1 + \frac{\beta}{2} \cdot (e^\varepsilon - 1)\right) \leq \ln\left(1 + \frac{\beta}{2} \cdot (2\varepsilon)\right) = \ln(1 + \beta\varepsilon) \leq \ln(e^{\beta\varepsilon}) = \varepsilon\beta,$$

where the inequalities hold because $e^\varepsilon - 1 \leq 2\varepsilon$ for all $\varepsilon \leq 1$ and $1 + \varepsilon\beta \leq e^{\varepsilon\beta}$ for all $\varepsilon\beta$.

Next, we argue that if sampler $\mathcal{A}$ is $\alpha$-accurate on class $\mathcal{C}$ for datasets of size $\mathsf{Po}(n)$, then sampler $\mathcal{A}_{\beta/2}$ is $\alpha$-accurate on class $\mathcal{C}$ for datasets of size $\mathsf{Po}(n \cdot \frac{2}{\beta})$. Suppose $\mathcal{A}_{\beta/2}$ is given a sample $\mathbf{X}$ of size $\mathsf{Po}\left(n \cdot \frac{2}{\beta}\right)$ drawn i.i.d. from some distribution $P$. Then, the size of $\mathbf{X}^*$, obtained by subsampling each entry of $\mathbf{X}$ with probability $\beta/2$, has distribution $\mathsf{Po}(n)$, and entries of $\mathbf{X}^*$ are i.i.d. from $P$. Since the output distributions of $\mathcal{A}_{\beta/2}(\mathbf{X})$ and $\mathcal{A}(\mathbf{X}^*)$ are the same, the accuracy guarantee is the same for both algorithms. $\qquad\square$

## B.3 General Samplers to Frequency-Count-Based Samplers

Our final transformation shows that algorithms that sample form distribution classes with certain symmetries can be assumed without loss of generality to use only frequency counts of their input dataset in their decisions. Before stating this result (Lemma B.7), we define *frequency counts*, *frequency-count-based algorithms*, and the type of symmetries relevant for the transformation.

**Definition B.4** (Frequency Counts). *Given a dataset $\mathbf{x}$ and an integer $j \geq 0$, let $F_j(\mathbf{x})$ denote the number of elements that occur $j$ times in $\mathbf{x}$. The vector $F(\mathbf{x})$ of* frequency counts *of a dataset $\mathbf{x}$ of size $n$ is $(F_0(\mathbf{x}), \ldots, F_n(\mathbf{x}))$.*

**Definition B.5** (Frequency-count-based algorithms). *A sampler is* frequency-count-based *if, for every element $i$ in the universe, the probability that the algorithm outputs $i$ when given a dataset $\mathbf{x}$ only depends on $j$, the number of occurrences of $i$ in $\mathbf{x}$, and on $F(\mathbf{x})$. If $\mathbf{x}$ contains an element $i \in \mathcal{U}$ that occurs $j$ times in $\mathbf{x}$, then let $p_{j, F(\mathbf{x})}$ denote the probability that the sampler outputs $i$; otherwise, let $p_{j, F(x)} = 0$.*

Next, we define the type of distribution classes for which our transformation works.

**Definition B.6.** *A class $\mathcal{C}$ of distributions over a universe $\mathcal{U}$ is* label-invariant *if for every distribution $P \in \mathcal{C}$ and permutation $\pi : \mathcal{U} \to \mathcal{U}$, we have $\pi(P) \in \mathcal{C}$, where $\pi(P)$ is the distribution obtained by applying permutation $\pi$ to the support of $P$ (that is, $\Pr_{\pi(P)}(s) = \Pr_P(\pi^{-1}(s))$ for all $s \in \mathcal{U}$).*

Examples of label-invariant classes include the class of all Bernoulli distibutions and, more generally, the class of all $k$-ary distributions, for any $k$.

**Lemma B.7.** *Fix a label-invariant distribution class $\mathcal{C}$. If there exists an $(\varepsilon, \delta)$-differentially private sampler $\mathcal{A}$ that is $\alpha$-accurate on $\mathcal{C}$ with a particular distribution on the dataset size, then there exists an $(\varepsilon, \delta)$-differentially private frequency-count-based sampler $\mathcal{A}_{FP}$ that is $\alpha$-accurate on $\mathcal{C}$ with the same distribution on the dataset size.*

*Proof.* Consider an $\alpha$-accurate sampler $\mathcal{A}$ for the class $\mathcal{C}$ that is not frequency-count-based. Construct the sampler $\mathcal{A}_{FP}$ as show in Algorithm 2.

---

**Algorithm 2** Sampler $\mathcal{A}_{FP}$

    **Input:** dataset $\mathbf{x}$, universe $\mathcal{U}$
    **Output:** $i \in \mathcal{U}$
1: Choose a permutation $\pi : \mathcal{U} \to \mathcal{U}$ uniformly at random.
2: **return** $\pi^{-1}(\mathcal{A}(\pi(\mathbf{x})))$

---

First, we show that $\mathcal{A}_{FP}$ is $\alpha$-accurate for $\mathcal{C}$. For all $P \in \mathcal{C}$ and $\mathbf{x} \sim P$, denote by $Q_{\mathcal{A}_{FP}(\mathbf{x})}$ the distribution of outputs for $\mathcal{A}_{FP}(\mathbf{x})$. Then, for a fixed permutation $\pi$, we have

$$
\begin{aligned}
d_{TV}(Q_{\mathcal{A}_{FP}(\mathbf{x})}, P) &= d_{TV}\left(\pi^{-1}(Q_{\mathcal{A}_{FP}(\pi(\mathbf{x}))}, P\right) \\
&= d_{TV}\left(Q_{\mathcal{A}_{FP}(\pi(\mathbf{x}))}, \pi(P)\right) \leq \alpha.
\end{aligned}
\tag{2}
$$

The equalities hold by the definition of $\mathcal{A}_{FP}$ and $\mathcal{A}$, and the inequality holds because $\pi(\mathbf{x}) \sim \pi(P)$ and since $\mathcal{C}$ is label-invariant. For a uniformly chosen $\pi$,

$$
\begin{aligned}
d_{TV}(Q_{\mathcal{A}_{FP}(\mathbf{x})}, P) &= d_{TV}\left(\mathbb{E}_{\pi}[Q_{\mathcal{A}_{FP}(\mathbf{x})|\pi}], P\right) \\
&\leq \mathbb{E}_{\pi}[d_{TV}(Q_{\mathcal{A}_{FP}(\mathbf{x})|\pi}, P)] \qquad \text{By the triangle inequality} \\
&\leq \max_{\pi}\{d_{TV}(Q_{\mathcal{A}_{FP}(\mathbf{x})|\pi}, P)\} \\
&= \max_{\pi}\{d_{TV}(Q_{\mathcal{A}_{FP}(\pi(\mathbf{x}))}, \pi(P))\} \leq \alpha. \qquad \text{By (2)}
\end{aligned}
$$

So, Algorithm $\mathcal{A}_{FP}$ is also $\alpha$-accurate for $\mathcal{C}$.

Next, we show that $\mathcal{A}_{FP}$ is frequency-count-based by proving that for all permutations $\pi^*$ on the universe and for all $i$ in the universe, $\Pr[\mathcal{A}_{FP}(\pi^*(\mathbf{x})) = \pi^*(i)] = \Pr[\mathcal{A}_{FP}(\mathbf{x}) = i]$. Let $\pi_0 = \pi \circ \pi^*$. We can characterise the output distribution of $\mathcal{A}_{FP}$ for a fixed $\mathbf{x}$ as follows

$$
\begin{aligned}
\Pr[\mathcal{A}_{FP}(\pi^*(\mathbf{x})) = \pi^*(i)] &= \frac{1}{|\mathcal{U}|!} \sum_{\pi_0 \in [\mathcal{U}!]} \Pr[\mathcal{A}(\pi_0 \circ \pi^*(\mathbf{x})) = \pi_0 \circ \pi^*(i)] \\
&= \frac{1}{|\mathcal{U}|!} \sum_{\pi_0 \circ \pi^* \in [\mathcal{U}!]} \Pr[\mathcal{A}(\pi_0 \circ \pi^*(\mathbf{x})) = \pi_0 \circ \pi^*(i)] \\
&= \frac{1}{|\mathcal{U}|!} \sum_{\pi \in [\mathcal{U}!]} \Pr[\mathcal{A}(\pi(\mathbf{x})) = \pi(i)] \\
&= \frac{1}{|\mathcal{U}|!} \sum_{\pi \in [\mathcal{U}!]} \Pr[\pi^{-1}(\mathcal{A}(\pi(\mathbf{x}))) = i] \\
&= \Pr[\mathcal{A}_{FP}(\mathbf{x}) = i]
\end{aligned}
$$

The third equality holds since the permutations $\pi, \pi^*$ are bijections. Thus $\mathcal{A}_{\text{FP}}$ is frequency-count-based.

Furthermore, the sizes of the input datasets to $\mathcal{A}_{\text{FP}}$ and $\mathcal{A}$ are identical, so if $\mathcal{A}$ takes a sample with size distributed according to some distribution $P$, then so does the frequency-count-based sampler $\mathcal{A}_{\text{FP}}$. $\qquad\square$

# C   $k$-ary Discrete Distributions

We consider privately sampling from the class of discrete distribution over $[k] := \{1, 2, \ldots, k\}$. We call this class $\mathcal{C}_k$. We prove in this section that the sample complexity of this task is $\Theta(k/\alpha\varepsilon)$, corresponding to Theorems 1.4 and 1.5. The proof of Theorem 1.5 is split into two cases: Theorem C.1 deals with the case where $k = 2$ and Theorem C.8 deals with the case where $k \geq 3$. We combine these theorems appropriately at the end of Section C.3.2.

## C.1   Optimal Private Sampler for $k$-ary Distributions

In this section, we prove Theorem 1.4.

*Proof of Theorem 1.4.* Algorithm 3 is the desired $(\varepsilon, 0)$-differentially private sampler for $\mathcal{C}_k$. The algorithm computes the empirical distribution, adds Laplace noise to each count in $[k]$, and then projects the result onto $\mathcal{C}_k$ in order to sample from the resulting distribution. The $L_1$ projection onto $\mathcal{C}_k$ is defined as $L_1 Proj(P) = \arg\min_{P' \in \mathcal{C}_k} \|P - P'\|_1$.

---

**Algorithm 3** Sampler $\mathcal{A}_{\mathcal{C}_k}$ for $\mathcal{C}_k$

---

    **Input:** dataset $\mathbf{x} \in [k]^n$, parameter $\varepsilon > 0$
    **Output:** $i \in [k]$
1: **for** $j = 1$ to $k$ **do**
2:     $\hat{P}_j \leftarrow \frac{1}{n} \sum_{i=1}^n \mathbb{1}_{[x_i = j]}$                           ▷ Compute the empirical distribution
3:     $Z_j \sim \mathsf{Lap}(2/\varepsilon n)$                                     ▷ Sample Laplace noise
4:     $\hat{P}_j^{noisy} \leftarrow \hat{P}_j + Z_j$                           ▷ Compute noisy empirical estimate
5: $\tilde{P} \leftarrow L_1 Proj(\hat{P}^{noisy})$            ▷ Do $L_1$ projection of private empirical estimate to $\mathcal{C}_k$
6: $i \sim \tilde{P}$                                          ▷ Sample from resulting distribution
7: **return** $i$

---

First, we argue that Algorithm 3 is $\alpha$-accurate. Let $P$ be the input distribution represented by a vector of length $k$. As defined in Algorithm 3, let $\hat{P}$ be the empirical distribution, $\hat{P}^{noisy}$ be the empirical distribution with added Laplace noise, and $\tilde{P}$ be the distribution obtained after applying $L_1$ projection (all represented by vectors of length $k$). Then $\mathbb{E}_{\mathbf{X}}[\hat{P}] = P$, since $\hat{P}$ is the empirical distribution of a dataset sampled from $P$. Let $Q_{\mathcal{A}_{\mathcal{C}_k}, P}$ be the distribution of the sampler's output for dataset $\mathbf{X} \sim P^n$. Then $Q_{\mathcal{A}_{\mathcal{C}_k}, P} = \mathbb{E}_{\mathbf{X}, \mathcal{A}}[\tilde{P}]$, since the output of $\mathcal{A}_{\mathcal{C}_k}$ is sampled from $\tilde{P}$. We get

$$d_{TV}(Q_{\mathcal{A}_{\mathcal{C}_k}, P}, P) = \frac{1}{2}\left\|P - Q_{\mathcal{A}_{\mathcal{C}_k}, P}\right\|_1 = \frac{1}{2}\left\|P - \mathop{\mathbb{E}}_{\mathbf{X}, \mathcal{A}_{\mathcal{C}_k}}[\tilde{P}]\right\|_1 = \frac{1}{2}\left\|\mathop{\mathbb{E}}_{\mathbf{X}, \mathcal{A}_{\mathcal{C}_k}}[\hat{P} - \tilde{P}]\right\|_1$$

$$\leq \frac{1}{2} \cdot \mathop{\mathbb{E}}_{\mathbf{X}, \mathcal{A}_{\mathcal{C}_k}}\left[\|\hat{P} - \tilde{P}\|_1\right], \tag{3}$$

where we applied Jensen's inequality in the last step of the derivation. Additionally,

$$\|\hat{P} - \tilde{P}\|_1 = \|\tilde{P} - \hat{P}^{noisy} + \hat{P}^{noisy} - \hat{P}\|_1$$
$$\leq \|\hat{P} - \hat{P}^{noisy}\|_1 + \|\tilde{P} - \hat{P}^{noisy}\|_1 \qquad \text{By the triangle inequality}$$
$$\leq 2\|\hat{P} - \hat{P}^{noisy}\|_1. \qquad\qquad\qquad \text{Definition of } L_1 \text{ projection}$$

Substituting this into (3), we get that

$$d_{TV}(P, Q_{\mathcal{A}_{\mathcal{C}_k}, P}) \leq \frac{1}{2} \cdot \mathop{\mathbb{E}}_{\mathbf{x}, \mathcal{A}_{\mathcal{C}_k}} \left[ 2\|\hat{P} - \hat{P}^{noisy}\|_1 \right] = \mathop{\mathbb{E}}_{\mathcal{A}_{\mathcal{C}_k}} \left[ \sum_{j \in [k]} |Z_j| \right] = \frac{2k}{n\varepsilon},$$

since the expectation of the absolute value of a random variable distributed according to the Laplace distribution $\mathsf{Lap}(\frac{2}{n\varepsilon})$ is $\frac{2}{n\varepsilon}$. We conclude that with $n \geq \frac{2k}{\alpha\varepsilon}$, Algorithm 3 is $\alpha$-accurate.

Next, we show that Algorithm 3 is $(\varepsilon, 0)$-differentially private. The sensitivity of a function $f : \mathcal{X}^n \to \mathbb{R}^d$ is defined as $\max_{\mathbf{x}, \mathbf{x}' \in \mathcal{X}^n, \|\mathbf{x} - \mathbf{x}'\|_0 = 1} \|f(\mathbf{x}) - f(\mathbf{x}')\|_1$. Recall that $\hat{P}$ is the empirical distribution of dataset $\mathbf{x}$ (represented by a vector of length $k$). Changing one element of $\mathbf{x}$ can change only two components of $\hat{P}$ by $\frac{1}{n}$ each. Hence, the sensitivity of the empirical distribution is $\frac{2}{n}$. Algorithm 3 adds Laplace noise scaled to the sensitivity of the empirical distribution to each component of the empirical distribution and then post-processes the output. This is an instantiation of the Laplace Mechanism (proved in [17] to be $(\varepsilon, 0)$-differentially private) followed by post-processing. Algorithm 3 is $(\varepsilon, 0)$-differentially private since differential privacy is preserved under post-processing. □

## C.2 The Lower Bound for the Class of Bernoulli Distributions

We consider the class $\mathcal{B}$ of Bernoulli distributions with an unknown bias $p$. For all $p \in [0, 1]$, distribution $\mathsf{Ber}(p) \in \mathcal{B}$ outputs 1 with probability $p$ and 0 with probability $1 - p$. Algorithm 3 for the special case of $k = 2$ shows that $O(\frac{1}{\alpha\varepsilon})$ samples are sufficient for $(\varepsilon, 0)$-differentially private $\alpha$-accurate sampling from $\mathcal{B}$. In this section, we show that this bound is tight, even for $(\varepsilon, \delta)$-differentially private samplers.

**Theorem C.1.** *If $\varepsilon \in (0, 1], \alpha \in (0, 1)$, and $\delta \leq \alpha\varepsilon$, then every $(\varepsilon, \delta)$-differentially private sampler that is $\alpha$-accurate on the class $\mathcal{B}$ of Bernoulli distributions requires $\Omega(\frac{1}{\alpha\varepsilon})$ samples.*

*Proof.* The following lemma captures how differential privacy affects a sampler for Bernoulli distributions.

**Lemma C.2.** *Suppose $\delta \leq \alpha\varepsilon$. If sampler $\mathcal{A}$ is $(\varepsilon, \delta)$-differentially private and $\alpha$-accurate on the class $\mathcal{B}$ of Bernoulli distributions then, for all $t \in [n]$,*

$$\Pr[\mathcal{A}(1^t 0^{n-t}) = 1] \leq 2\alpha e^{\varepsilon t}.$$

*Proof.* Fix $n$ and $t \in [n]$. Since $\mathcal{A}$ is $\alpha$-accurate on $\mathsf{Ber}(0)$, we have $\Pr[\mathcal{A}(0^n) = 1] \leq \alpha$. We start with the dataset $1^t 0^{n-t}$ and replace 1s with 0s one character at a time until we reach $0^n$. Since $\mathcal{A}$ is $(\varepsilon, \delta)$-differentially private, its output distribution does not change dramatically with every replacement. Specifically,

$$\begin{aligned}
\Pr[\mathcal{A}(1^t 0^{n-t}) = 1] &\leq e^\varepsilon \cdot \Pr[\mathcal{A}(1^{t-1} 0^{n-t+1}) = 1] + \delta \\
&\leq e^\varepsilon (e^\varepsilon \cdot \Pr[\mathcal{A}(1^{t-2} 0^{n-t+2}) = 1] + \delta) + \delta \leq \ldots \\
&\leq e^{\varepsilon t} \cdot \Pr[\mathcal{A}(0^n) = 1] + \delta \cdot \sum_{i=0}^{t-1} e^{\varepsilon t} = e^{\varepsilon t} \cdot \Pr[\mathcal{A}(0^n) = 1] + \delta \cdot \frac{e^{\varepsilon t} - 1}{e^\varepsilon - 1} \\
&\leq e^{\varepsilon t} \cdot \alpha + \delta \cdot \frac{e^{\varepsilon t} - 1}{e^\varepsilon - 1} \leq e^{\varepsilon t} \left( \alpha + \frac{\delta}{\varepsilon} \right) \leq 2\alpha e^{\varepsilon t},
\end{aligned}$$

where the last two inequalities hold because $e^\varepsilon - 1 \leq \varepsilon$ for all $\varepsilon$ and since $\delta \leq \alpha\varepsilon$. □

Consider a sampler $\mathcal{A}$, as described in Theorem C.1. By Lemma B.7, since the class $\mathcal{B}$ is label-invariant, we may assume w.l.o.g. that $\mathcal{A}$ is frequency-count-based. In particular, the output distribution of $\mathcal{A}$ is the same on datasets with the same number of 0s and 1s.

Consider a Bernoulli distribution with $p = 10\alpha$. Let $T$ be a random variable that denotes the number of 1s in $n$ independent draws from $\mathsf{Ber}(p)$. Then $T$ has binomial distribution $Bin(n, 10\alpha)$. By Claim A.20 and

$\alpha$-accuracy of $\mathcal{A}$ for $\mathsf{Ber}(p)$, we get

$$9\alpha \leq \mathop{\mathbb{E}}_{\mathbf{X} \sim (\mathsf{Ber}(p))^{\otimes n}}[\mathcal{A}(X) = 1] = \mathop{\mathbb{E}}_{T \sim Bin(n,10\alpha)}[\mathcal{A}(1^T 0^{n-T}) = 1]$$

$$\leq \mathop{\mathbb{E}}_{T \sim Bin(n,10\alpha)}[2\alpha e^{\varepsilon T}] = 2\alpha(10\alpha(e^\varepsilon - 1) + 1)^n \tag{4}$$

$$\leq 2\alpha(20\alpha\varepsilon + 1)^n \leq 2\alpha e^{20\alpha\varepsilon n}, \tag{5}$$

where, to get (4), we used Lemma C.2 and then the moment generating function of the binomial distribution; in (5), we used that $e^\varepsilon - 1 \leq 2\varepsilon$ for all $\varepsilon \in (0, 1]$ and, finally, that $x + 1 \leq e^x$ for all $x$ (applied with $x = 20\alpha\varepsilon$). We obtained that $9\alpha \leq 2\alpha \cdot e^{200\alpha\varepsilon n}$, so $n \geq \frac{20}{\ln 4.5} \frac{1}{\alpha\varepsilon}$ samples are required. This completes the proof of Theorem C.1. $\qquad\square$

## C.3   The Lower Bound for the Class of $k$-ary Distributions

In this section, we prove Theorem 1.5 by proving a lower bound for the universe size at least 3 (Theorem C.8) and combining it with the previously proved lower bound for the binary case (Theorem C.1). The crux of the proof of Theorem C.8 is presented in Section C.3.1, where we state and prove the lower bound for the special case of Poisson, frequency-count-based samplers with sufficiently small $\varepsilon$ (that is, a strong privacy guarantee). In Section C.3.2, we complete the proof of the theorem by generalizing the lower bound from Section C.3.1 with the help of the transformation lemmas (Lemmas B.1, B.3, and B.7) that allow us to convert general samplers to Poisson, frequency-count-based algorithms with small privacy parameter $\varepsilon$.

### C.3.1   The Lower Bound for Poisson, Frequency-Count-Based Samplers with Small $\varepsilon$

**Lemma C.3.** *Fix $k, n \in \mathbb{N}, \alpha \in (0, 0.02], \varepsilon \in (0, 1/\ln(1/\alpha)]$, and $\delta \in [0, 0.1\alpha\varepsilon/k]$. Let $\mathcal{C}_{2k+1}$ denote the class of discrete distributions over the universe $[2k+1]$. If sampler $\mathcal{A}$ is $(\varepsilon, \delta)$-differentially private, frequency-count-based, and $\alpha$-accurate on class $\mathcal{C}_{2k+1}$ with dataset size distributed as $\mathsf{Po}(n)$, then $n > \frac{1}{60} \cdot \frac{k}{\alpha\varepsilon}$.*

*Proof.* We consider the following distribution $P$ over the universe $\mathcal{U} = [2k + 1]$. Fix $\alpha^* = 60\alpha$ and a set $S^* \subset [2k]$ of size $k$. Distribution $P$ has mass $\alpha^*/k$ on each element in $S^*$ and mass $1 - \alpha^*$ on the *special* element $2k + 1$.

Consider a sampler $\mathcal{A}$ satisfying the conditions of Lemma C.3. Let $Q_{\mathcal{A},P}$ denote the output distribution of $\mathcal{A}$ when the dataset size $N \sim \mathsf{Po}(n)$ and the dataset $\mathbf{X} \sim P^{\otimes N}$. Observe that

$$d_{TV}(Q_{\mathcal{A},P}, P) \geq \Pr_{\substack{N \sim \mathsf{Po}(n) \\ \mathbf{X} \sim P^{\otimes N}}}[\mathcal{A}(\mathbf{X}) \notin Supp(P)]. \tag{6}$$

We will show that when $n \leq \frac{k}{60\alpha\varepsilon}$ and $\varepsilon$ and $\delta$ are in the specified range, the right-hand side of (6) is large.

We start by deriving a lower bound on $\Pr[\mathcal{A}(\mathbf{x}) \notin Supp(P)]$ for a fixed dataset $\mathbf{x}$ of a fixed size $N$. Since $\mathcal{A}$ is frequency-count-based, the probability that it outputs a specific element in $[2k]$ that occurs 0 times in $\mathbf{x}$ is $p_{0,F(\mathbf{x})}$. Let $F_0^*(\mathbf{x})$ denote the number of elements in $[2k]$ that occur 0 times in $\mathbf{x}$ (note that the special element $2k + 1$ is excluded from this count). By definition, $F_0^*(\mathbf{x}) \leq 2k$. Consequently,

$$\Pr[\mathcal{A}(\mathbf{x}) \notin Supp(P)] = k \cdot p_{0,F(\mathbf{x})} \geq \frac{1}{2} \cdot F_0^*(\mathbf{x}) \cdot p_{0,F(\mathbf{x})}. \tag{7}$$

The next claim uses the fact that sampler $\mathcal{A}$ is $(\varepsilon, \delta)$-differentially private to show that the probability $p_{j,F(\mathbf{x})}$ (that $\mathcal{A}$ outputs some specific element in the universe $\mathcal{U}$ that appears $j$ times in the dataset $\mathbf{x}$) cannot be much larger than the probability that $\mathcal{A}$ outputs a specific element in $\mathcal{U}$ that does not appear in $\mathbf{x}$.

**Claim C.4.** *For every $(\varepsilon, \delta)$-differentially private sampler and every frequency count $f \in \mathbb{Z}^*$ and index $j \in \mathcal{U}$,*

$$p_{j,f} \leq e^{\varepsilon j}\left(p_{0,f} + \frac{\delta}{\varepsilon}\right). \tag{8}$$

*Proof.* Consider a frequency count $f$ and a dataset $\mathbf{x}$ with $F(\mathbf{x}) = f$. Note that (8) is true trivially for all $j$ such that $F_j(\mathbf{x}) = 0$ because, in that case, $p_{j,F(\mathbf{x})}$ is set to 0.

Fix any $j \in \mathcal{U}$ such that $F_j(\mathbf{x}) > 0$. Let $a$ be any element in $\mathcal{U}$ that occurs $j$ times in the dataset $\mathbf{x}$. Let $b$ be any element in $\mathcal{U}$ that is not in the support of the distribution $P$. Let $\mathbf{x}|_{a\to b}$ denote the dataset obtained by replacing every instance of $a$ in the dataset $\mathbf{x}$ with element $b$. By group privacy [17],

$$\Pr[\mathcal{A}(\mathbf{x}) = a] \le e^{j\varepsilon} \Pr[\mathcal{A}(\mathbf{x}|_{a\to b}) = a] + \delta \cdot \frac{e^{\varepsilon j} - 1}{e^{\varepsilon} - 1}. \tag{9}$$

Note that the dataset $\mathbf{x}|_{a\to b}$ does not contain element $a$, since we've replaced every instance of it with $b$. Importantly, $F(\mathbf{x}|_{a\to b}) = F(\mathbf{x})$ because $b$ is outside of the support of the distribution $P$ and hence does not occur in $\mathbf{x}$. Since $\mathcal{A}$ is frequency-count-based and $F(\mathbf{x}) = F(\mathbf{x}|_{a\to b})$, we get that $p_{0,F(\mathbf{x})} = p_{0,F(\mathbf{x}|_{a\to b})}$. Substituting this into (9) and using the fact that $e^{\varepsilon} - 1 \ge \varepsilon$ for all $\varepsilon$, we get that

$$p_{j,F(\mathbf{x})} \le e^{j\varepsilon} \cdot p_{0,F(\mathbf{x})} + \delta \cdot \frac{e^{\varepsilon j} - 1}{e^{\varepsilon} - 1} \le e^{\varepsilon j} \left( p_{0,f} + \frac{\delta}{\varepsilon} \right).$$

This completes the proof of Claim C.4. $\qquad\square$

For a dataset $\mathbf{x}$ and $i \in [2k + 1]$, let $N_i(\mathbf{x})$ denote the number of occurrences of element $i$ in $\mathbf{x}$. Next, we give a lower bound on $\Pr[\mathcal{A}(\mathbf{x}) \notin Supp(P)]$ in terms of the counts $N_i(\mathbf{x})$.

**Claim C.5.** *Let $N \in \mathbb{N}$ and $\mathbf{x} \in [2k + 1]^N$ be a fixed dataset. Set $Y = \sum_{i \in S^*} \left[ e^{N_i(\mathbf{x})\varepsilon} \right]$. Then*

$$\Pr[\mathcal{A}(\mathbf{x}) \notin Supp(P)] \ge \frac{1}{2} \cdot \frac{\Pr[\mathcal{A}(\mathbf{x}) \in [2k]]}{1 + Y/k} - \frac{k\delta}{\varepsilon}.$$

*Proof.* In the following derivation, we use the fact that that an element $j \in [2k]$ that appears $j$ times in $\mathbf{x}$ is returned by $\mathcal{A}$ with probability $p_{j,F(\mathbf{x})}$, then split the elements into those that do not appear in $\mathbf{x}$ and those that do, next use the fact that all elements from $[2k]$ that appear in $\mathbf{x}$ must be in $S^*$, then apply Claim C.4, and finally substitute $Y$ for $\sum_{i \in S^*} \left[ e^{N_i(\mathbf{x})\varepsilon} \right]$:

$$\begin{aligned}
\Pr[\mathcal{A}(\mathbf{x}) \in [2k]] = \sum_{i \in [2k]} p_{N_i(\mathbf{x}),F(\mathbf{x})} &= F_0^*(\mathbf{x}) \cdot p_{0,F(\mathbf{x})} + \sum_{i \in [2k] \cap \mathbf{x}} p_{N_i(\mathbf{x}),F(\mathbf{x})} \\
&\le F_0^*(\mathbf{x}) \cdot p_{0,F(\mathbf{x})} + \sum_{i \in S^*} p_{N_i(\mathbf{x}),F(\mathbf{x})} \\
&\le F_0^*(\mathbf{x}) \cdot p_{0,F(\mathbf{x})} + \sum_{i \in S^*} p_{0,F(\mathbf{x})} \cdot \left( e^{\varepsilon N_i(\mathbf{x})} + \frac{\delta}{\varepsilon} \right) \\
&\le \left( F_0^*(\mathbf{x}) + Y \right) \left( p_{0,F(\mathbf{x})} + \frac{\delta}{\varepsilon} \right).
\end{aligned}$$

We rearrange the terms to get

$$p_{0,F(\mathbf{x})} \ge \frac{\Pr[\mathcal{A}(\mathbf{x}) \in [2k]]}{F_0^*(\mathbf{x}) + Y} - \frac{\delta}{\varepsilon}.$$

Substituting this bound on $p_{0,F(\mathbf{x})}$ into (7), we obgain

$$\begin{aligned}
\Pr[\mathcal{A}(\mathbf{x}) \notin Supp(P)] &\ge \frac{1}{2} \cdot \frac{F_0^*(\mathbf{x}) \Pr[\mathcal{A}(\mathbf{x}) \in [2k]]}{F_0^*(\mathbf{x}) + Y} - \frac{1}{2} \cdot \frac{F_0^*(\mathbf{x}) \cdot \delta}{\varepsilon} \\
&= \frac{1}{2} \cdot \frac{\Pr[\mathcal{A}(\mathbf{x}) \in [2k]]}{1 + Y/F_0^*(\mathbf{x})} - \frac{1}{2} \cdot \frac{F_0^*(\mathbf{x}) \cdot \delta}{\varepsilon} \\
&\ge \frac{1}{2} \cdot \frac{\Pr[\mathcal{A}(\mathbf{x}) \in [2k]]}{1 + Y/k} - \frac{k\delta}{\varepsilon},
\end{aligned}$$

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

### C.3.2    Final Lower Bound for $k$-ary Distributions

In this section, we complete the proof of Theorem C.8.

**Theorem C.8.** *For all sufficiently small $\alpha > 0$, $k, n \in \mathbb{N}$, $\varepsilon \in (0, 1]$, , and $\delta \in \left[0, 0.1 \cdot \frac{\alpha\varepsilon}{k}\right]$, if there exists an $(\varepsilon, \delta)$-differentially private sampler that is $\alpha$-accurate on the class $\mathcal{C}_{2k+1}$ of discrete distributions over universe $[2k + 1]$ on datasets of size $n$, then $n = \Omega(\frac{k}{\alpha\varepsilon})$.*

*Proof.* We apply Lemmas B.1, B.3, and B.7 to generalize the lower bound in Lemma C.3 to work for all differentially private samplers and all privacy parameters $\varepsilon \in (0, 1]$.

Suppose there exists an $(\varepsilon, \delta)$-differentially private sampler $\mathcal{A}$ that is $\alpha$-accurate on the class $\mathcal{C}_{2k+1}$ for datasets of size $n$, for some $n \in \mathbb{N}, \varepsilon \in (0, 1], \delta \in \left[0, 0.1 \cdot \frac{\alpha\varepsilon}{k}\right]$, and $\alpha \in (0, 0.01]$. We can assume without loss of generality that $n \geq 6 \ln(1/\alpha)$, since, if this is not the case, $\mathcal{A}$ can ignore extra samples. By Lemma B.1, there exists an $(\varepsilon, \delta)$-differentially private sampler $\mathcal{A}_{\mathsf{Po}}$ that is $(\alpha + e^{-n/6})$-accurate on $\mathcal{C}_{2k+1}$ when its dataset size is distributed as $\mathsf{Po}(2n)$. Since $n \geq 6 \ln(1/\alpha)$, this gives $e^{-n/6} \leq \alpha$, so sampler $\mathcal{A}_{\mathsf{Po}}$ is $2\alpha$-accurate. By Lemma B.3, we can amplify the privacy to construct an $(\varepsilon', \delta')$-differentially private sampler $\mathcal{A}'$ that is $\alpha'$-accurate for datasets with size distributed as $\mathsf{Po}(4n \ln(1/\alpha'))$, where $\alpha' = 2\alpha$, $\varepsilon' = \frac{\varepsilon}{\ln(1/\alpha')}$, and $\delta' = \frac{\delta}{2\ln(1/\alpha')}$. Then $\varepsilon' \leq \frac{1}{\ln(1/\alpha')}, \delta' \leq 0.01 \frac{\alpha'\varepsilon'}{k}$, and $\alpha' \leq 0.02$, as required to apply Lemma C.3 with privacy parameters $\varepsilon', \delta'$ and accuracy parameter $\alpha'$. By Lemma B.7, we can assume the sampler is frequency-count-based with no changes in the privacy and accuracy parameters. Now, applying Lemma C.3 gives

$$4n \ln(1/\alpha') \geq \frac{1}{60} \cdot \frac{k}{2\alpha \cdot \frac{\varepsilon}{\ln(1/\alpha')}} .$$

Therefore, $n \geq \frac{k}{480\alpha\varepsilon} = \Omega(\frac{k}{\alpha\varepsilon})$, as claimed in the theorem statement. $\qquad\square$

Now we can combine Theorem C.1 (for $k = 2$) and Theorem C.8 (for $k \geq 3$) to get the lower bound in Theorem 1.5 for all $k \geq 2$. Note that directly applying these theorems would give a lower bound for $\delta \in [0, 0.1 \frac{\alpha\varepsilon}{k}]$. However, this can be extended to any $\delta \in [0, \frac{1}{5000n}]$. To see this, observe that when $n < \frac{k}{480\alpha\epsilon} < \frac{1}{5000\delta}$, a direct application of our theorems proves the lower bound. When $n < \frac{1}{5000\delta} < \frac{k}{480\alpha\varepsilon}$,

assume by way of contradiction that there exists an $(\epsilon, \delta)$-DP sampler that is $\alpha$-accurate on the class of $k$-ary distributions for input datasets of size $n$.

One can find a triple of values $(k', \alpha', \varepsilon')$ such that $2 \leq k' \leq k$, $0.01 \geq \alpha' \geq \alpha$ and $1 \geq \varepsilon' \geq \varepsilon$ such that $n \leq \frac{k'}{480\alpha'\varepsilon'} < \frac{1}{5000\delta}$. In particular, $\delta < 0.1 \cdot \frac{\alpha'\varepsilon'}{k'}$. Next, note that a sampler that achieves accuracy $\alpha < 0.01$ is also a sampler that achieves accuracy $\alpha'$ for all $0.01 \geq \alpha' > \alpha$. Additionally, since the class of discrete distributions over $[k']$ is a subclass of the class of discrete distributions over $[k]$, an $(\varepsilon, \delta)$-differentially private sampler that is $\alpha$-accurate on the class of discrete distributions over $[k]$ when given an input dataset of size $n$ is $(\varepsilon', \delta)$-differentially private and $\alpha'$-accurate on the class of discrete distributions over $[k']$ for the same dataset size $n$. We can then apply the lower bounds in either Theorem C.8 or Theorem C.1 to show that no such sampler exists. This proves the theorem for all $\delta \in [0, \frac{1}{5000n}]$.

# D    Product Distributions Over $\{0, 1\}^d$

In this section, we consider the problem of privately sampling from the class $\mathcal{B}^{\otimes d}$ of product distributions over $\{0, 1\}^d$. We present and analyze a $\rho$-zCDP sampler for $\mathcal{B}^{\otimes d}$ (Theorem D.1) and then apply a standard conversion from $\rho$-zCDP to $(\epsilon, \delta)$-differential privacy (Theorem A.4) to prove Theorem 1.6.

**Theorem D.1** (Upper bound for product distributions). *For all $\rho \in (0, 1]$, $\alpha \in (0, 1)$, and $d$ greater than some sufficiently large constant, there exists a $\rho$-zCDP sampler for the class $\mathcal{B}^{\otimes d}$ of product Bernoulli distributions that is $\alpha$-accurate on datasets of size $n = O\left( \frac{d}{\alpha\sqrt{\rho}} \cdot \left[ \log^{9/4} d + \log^{5/4} \frac{1}{\alpha\sqrt{\rho}} \right] \right)$.*

This theorem implies Theorem 1.6; we prove the implication at the end of this section.

Our main technical tool is the recursive preconditing technique of [19]. Let $\mathbf{p} = (p_1, \ldots, p_d)$ be the unknown attribute biases for the product distribution $P \in \mathcal{B}^{\otimes d}$ from which the data is drawn. For some intuition, consider the following natural differentially private algorithm for sampling from a product distribution: First, privately estimate each of the attribute biases $p_j$ by adding noise to the sample mean; then sample each attribute independently from a Bernoulli with this estimated bias. This approach does not work directly because the $\ell_2$-sensitivity of the vector of sample means is $\sqrt{d}/n$. To accurately estimate tiny biases, we require a large sample size $n$. For instance, in the case where all the attribute biases are roughly $1/d$, the naive algorithm described above would require $n = \Omega(d^{3/2})$ records to be $\alpha$-accurate for a small constant $\alpha$.

To get around this (in the context of distribution learning), Kamath et al. [19] observe that when the biases are small and the input is drawn from a product distribution, the number of 1s in each record is constant—say, at most 10—with high probability. Viewing the records as vectors in $\mathbb{R}^d$, we can therefore truncate every record so that its $\ell_2$-norm is at most $\sqrt{10}$ (that is, we leave short vectors alone and shrink longer records) and then average the truncated data entries to obtain a *truncated mean*. We call $\sqrt{10}$ the *truncation ceiling*. Truncation reduces sensitivity, which allows one add less noise—and thus give better attribute bias estimates—while preserving privacy. When the biases are at most $1/d$, the sample complexity for constant accuracy $\alpha$ is reduced to $O(d/\varepsilon)$. (A similar idea works for biases very close to 1. For simplicity, we assume that all attribute biases $p_i$ are between 0 and $1/2$. See Footnote 4.)

The challenge with this approach is that we don't know biases ahead of time; when coordinates have large bias, setting the truncation ceiling too low leads to high error. Kamath et al. address this by estimating the attribute biases in rounds: in round $j$, attributes with biases close to $2^{-j}$ are estimated reasonably accurately, while smaller biases are passed to the next round where truncation can be applied more aggressively. This process is called *recursive preconditioning*, and it is an important part of our algorithm.

Our algorithm proceeds in two phases.

- **Bucketing Phase:** This phase implements recursive private preconditoning from [19] to estimate the attribute biases $p_i$. The main difference is that, for coordinates with large bias, we require less accurate estimates than [19] and can thus use fewer samples.

  In a bit more detail: The interval $[0, \frac{1}{2}]$ is divided into $\lceil \log_2 d \rceil + 1$ overlapping sub-intervals that we call *buckets*. The $r^{th}$ bucket corresponds to the interval $[\frac{1}{4} \cdot 2^{-r}, \ 2^{-r}]$. The exception is the smallest bucket, which corresponds to $[0, \frac{1}{d}]$.

We proceed in rounds, one per bucket. The bucketing phase uses half of the overall dataset and, for simplicity, those records are split evenly among rounds. Each round thus uses $m \approx \frac{n}{2 \log d}$ records. At round $r$, some coordinates are classified as having attributes in bucket $r$, while others are passed to the next round. With high probability, we maintain the invariants that (a) *only* coordinates with bias at most $2^{-r}$ are passed to round $r$, and (b) *all* coordinates with bias at most $2^{-r-2}$ are passed to round $r+1$ (except for the last round, in which no records are passed on). As a result, coordinates classified in round $r$ have biases in the bucket $[2^{-r-2}, 2^r]$; records left in the last round have bias at most $1/d$.

For example, the first round corresponds to bucket $[\frac{1}{8}, \frac{1}{2}]$. All coordinates are passed to that round (they have bias at most $\frac{1}{2}$ by assumption). Using its batch of $m$ records, this round of the algorithm computes the empirical means for all coordinates, adds Gaussian noise about $\frac{\sqrt{d}}{m}$ to each, and releases the list of noisy means. We select $n$ large enough for these noisy estimates to each be within $\frac{1}{16}$ of the true attribute bias with high probability (over the sampling of both the data and the noise). Attributes with noisy estimates below $3/16$ are passed to round 2, while the rest are assigned to bucket 1. One can check that the invariants are maintained: attributes with bias below $1/8$ are passed to round 2; those with bias at least $1/4$ are assigned to bucket 1; and those in between may go either way.[4]

At round $r$, we proceed similarly except that we can restrict records to those attributes that were passed to this round and we can truncate records so their $\ell_2$ norm is at most $T_r \approx \sqrt{d}/2^r$. When the data are from a product distribution and prior rounds were correct, this truncation has essentially no effect on the records but allows us to add less noise. We get noisy means that are within $\pm \frac{1}{8} \cdot 2^{-r}$ of the true biases. The invariants are maintained if we pass biases with noisy means below $\frac{3}{8} \cdot 2^{-r}$ to the next round, and assign the rest to bucket $r$.

- **Sampling Phase:** In the second phase, we use fresh data for the sampling phase to construct new, *unbiased* noisy empirical estimates of the attribute biases. In round $r$ of this phase, we restrict records to the attributes assigned to bucket $r$. We can truncate the records to have norm $T_r$ (because the biases are at most $2^{-r}$) and add noise as before. This gives us a list of noisy means, which we clip to $[0, 1]$ by rounding up negative values and rounding down values above 1. We sample one bit for each attribute independently, according to these clipped noisy means.

  For attributes in all buckets except the last, we get noisy means that lie in $[0, 1]$ with high probability (because the biases are at least $\frac{1}{4} \cdot 2^{-r}$). Since the estimates are unbiased and no clipping occurs, we sample from the correct distribution. For the attributes in the last bucket, we may get negative noisy means. However, the noise is small in these attributes, and we can bound the overall effect on the distribution. Interestingly, almost all the error of our algorithm comes from these low-bias attributes.

We present our sampler $\mathcal{A}_{prod}$ for $\mathcal{B}^{\otimes d}$ in Algorithm 4. Let $\mathbf{x} = (x_1, \ldots, x_n)$ be a dataset with $n$ records. The truncated mean operation, used in the algorithm, is defined as follows:

$$trunc_B(x_i) = \begin{cases} x_i & \text{if } \|x_i\|_2 \le B; \\ \frac{B}{\|x_i\|_2} x_i & \text{otherwise}; \end{cases}$$

$$tmean_B(\mathbf{x}) = \frac{1}{n} \sum_{i=1}^{n} trunc_B(x_i).$$

Recall that we assume that all of the attribute biases $p_j \in [0, 1/2]$.

First, we argue that this algorithm is private.

**Lemma D.2.** $\mathcal{A}_{prod}$ *is $\rho$-zCDP.*

---

[4] One can also handle biases larger than $1/2$ at this phase. Specifically, the first round of noisy measurements allows us to divide the coordinates into three disjoint sets, each containing only coordinates with biases in $[0, 1/4]$, $[1/8, 7/8]$, and $[3/4, 0]$, respectively. We can work with the coordinates in the first two sets as they are. For coordinates in the third set, we can flip all entries (from 0 to 1 and vice-versa), treat them as if their biases were in $[0, 1/4]$, and flip the corresponding output bits.

---

**Algorithm 4** Sampler $\mathcal{A}_{prod}$ for $\mathcal{B}^{\otimes d}$

---

**Input:** dataset $\mathbf{x} \in \{0,1\}^{d \times n}$, privacy parameter $\rho \in (0,1]$, failure parameter $\beta > 0$.
**Output:** $b \in \{0,1\}^d$

1: Set $R \leftarrow \log_2 \frac{d}{40}, m \leftarrow \frac{n}{2R+1}$.         $\triangleright$ For analysis, assume $m = \frac{1200d}{\alpha\sqrt{2\rho}} \log^{5/4} \frac{dR}{\alpha\beta\sqrt{2\rho}}$
2: Split $\mathbf{x}$ into $2R+1$ datasets $\mathbf{x}^1, \ldots, \mathbf{x}^{2R+1}$ of size $m$.

     **Bucketing Phase**
3: Set $S_1 \leftarrow [d], u_1 \leftarrow \frac{1}{2}, \tau_1 \leftarrow \frac{3}{16}$.
4: **for** $r = 1$ to $R$ **do**
5:      $S_{r+1} \leftarrow \emptyset, T_r \leftarrow \sqrt{6u_r|S_r|\log\frac{mR}{\beta}}$.
6:      Set $\tilde{p}[S_r] \leftarrow \text{tmean}_{T_r}(\mathbf{x}^r[S_r]) + \mathcal{N}(0, \frac{T_r^2}{2\rho m^2}\mathbb{I})$.         $\triangleright$ Form noisy bias estimates
7:      **for** $j \in S_r$ **do**
8:         **if** $\tilde{p}[j] < \tau_r$ **then**         $\triangleright$ Compare noisy bias estimate to threshold
9:             $S_{r+1} \leftarrow S_{r+1} \cup \{j\}$.         $\triangleright$ Send $j$ to next round.
10:     $S_{R+r} \leftarrow S_r \setminus S_{r+1}, T_{R+r} \leftarrow T_r$.
11:     $\tau_{r+1} \leftarrow \frac{\tau_r}{2}, u_{r+1} \leftarrow \frac{u_r}{2}$.

     **Sampling Phase**
12: $T_{2R+1} \leftarrow \sqrt{200\log\frac{m}{\beta}}, S_{2R+1} \leftarrow S_{R+1}, S_{R+1} \leftarrow S_1 \setminus S_2$.
13: **for** $r = R+1$ to $2R+1$ **do**
14:     $\tilde{p}[S_r] \leftarrow \text{tmean}_{T_r}(\mathbf{x}^r[S_r]) + \mathcal{N}(0, \frac{T_r^2}{2\rho m^2}\mathbb{I})$      $\triangleright$ Estimate biases using fresh data and noise
15:     **for** $j \in S_r$, **do**
16:        Set $q[j] \leftarrow [\tilde{p}[j]]_0^1$.         $\triangleright$ Clip to lie in the interval [0,1]
17:        Sample $b_j \sim Ber(q[j])$.         $\triangleright$ Sample from estimated marginal distribution.
18: **return** $(b_1, \ldots, b_d)$.

---

*Proof.* Each input record $x_i$ is used only in one round in one phase. Assume without loss of generality that this round is in the bucketing phase. The $\ell_2$-sensitivity of the truncated mean $tmean_{T_r}(\mathbf{X}^r[S_r])$ is $T_r/m$. By the privacy of the Gaussian mechanism (Lemma A.13), the step that produces this estimate is $\rho$-zCDP. The remaining steps simply post-process this estimate. Hence, by Lemma A.5, Algorithm 4 is $\rho$-zCDP. $\quad\square$

## D.1   Overview of Accuracy Analysis

We analyze the two phases of Algorithm 4 separately. Our analysis of the bucketing phase mirrors that of [19]. (Their results are not directly applicable to our setting because our algorithm use fewer samples. We therefore give new lemma statements and proofs.)

     In Section D.2, we prove technical lemmas that are used multiple times in the analysis of both phases. In Section D.3, we show that with high probability, the bucketing phase is successful—that is, we classify all of the attribute biases into the right buckets. This is encapsulated by Lemma D.7. This corresponds to obtaining good multiplicative approximations of all attribute biases except the smallest ones, for which we obtain good additive approximations.

     Next, in Section D.4, we prove a key lemma regarding the success of the sampling phase.

     The intuition behind the analysis of this phase is as follows. Algorithm $\mathcal{A}_{prod}$ samples its output from a product distribution. Since the input distribution is in $\mathcal{B}^{\otimes d}$, each attribute of an input record is sampled from a Bernoulli distribution. By the subadditivity of total variation distance for product distributions, the overall accuracy of $\mathcal{A}_{prod}$ can be bounded by showing that $\mathcal{A}_{prod}$ samples each attribute independently from a Bernoulli distribution with bias close to the true attribute bias $p_j$.

     The main idea is that the empirical attribute bias has expectation equal to the true attribute bias. Additionally, to preserve privacy, we add zero-mean Gaussian noise. Hence, a noisy empirical estimate of

the true attribute bias has mean equal to that attribute bias. If we knew for sure that the noisy empirical estimate for an attribute bias in the sampling phase was always between 0 and 1, then the sampler would sample this attribute from exactly the right Bernoulli distribution.

Alas, the noisy empirical estimate of an attribute bias could be less than 0 or larger than 1, and we would have to clip it to the interval $[0, 1]$ before sampling. This clipping introduces error since we no longer necessarily sample from the right distribution in expectation. We get around this by proving that for attribute biases $p_j$ larger than $\frac{\alpha}{d}$, clipping happens with low probability, and hence the loss in accuracy caused by clipping is small in expectation. However, for attribute biases $p_j$ that are smaller than $\frac{\alpha}{d}$, clipping could occur with high probability. For such attribute biases, we argue that the absolute difference between the clipped noisy empirical mean estimates and the true attribute biases is small enough (at most $\frac{\alpha}{d}$) with high probability. This argument is described in Lemma D.8.

Finally, we prove the main upper bound theorem in Section D.5 by putting everything together.

## D.2 Analysis of Good Events

In the accuracy analysis, we assume that $m \geq \frac{1200\,d}{\alpha\sqrt{2\rho}} \cdot \log^{5/4}\left(\frac{dR}{\alpha\beta\sqrt{2\rho}}\right)$ and $d$ is sufficiently large (that is, greater than some positive constant). In this section, we define three good events $G_1, G_2$, and $G_3$ that, respectively, represent that empirical means are close to the attribute biases in all $2R + 1$ datasets into which Algorithm 4 subdivides its input, that truncation does not occur in any round (assuming successful bucketing for that round), and that the added Gaussian noise is sufficiently small. We show that each of these events fails to occur only with small probability.

First, we prove that the empirical means are close to the attribute biases with high probability. Define the empirical mean $\hat{p}_r[j] := \frac{1}{m} \sum_{i=1}^{m} x_i^r[j]$.

**Lemma D.3.** *Let $G_1$ be the event that for all rounds $r \in [2R + 1]$, the following conditions hold:*

1. *For all $j \in [d]$, if $\frac{\alpha}{d} \leq p_j \leq \frac{1}{2}$ then $|\hat{p}_r[j] - p_j| \leq \frac{p_j}{16}$.*

2. *For all $j \in [d]$, if $p_j < \frac{\alpha}{d}$ then $|\hat{p}_r[j] - p_j| \leq \frac{\alpha}{4d}$.*

*Then, $\Pr\left[\overline{G_1}\right] \leq 2\beta$, where the probability is over the randomness of the input data and the coins of $\mathcal{A}_{prod}$.*

*Proof.* Fix $r \in [2R + 1]$ and $j \in [d]$. Note that $\mathbb{E}[\hat{p}_r[j]] = p_j$ for all $r \in [2R + 1]$.

We prove Item 1 of the lemma by a case analysis on $p_j$. First, when $p_j \geq \frac{\alpha}{4d}$, we use the multiplicative Chernoff bound from Claim F.2 for $\gamma \in (0, 1)$:

$$\Pr\left[\hat{p}_r[j] > p_j\left(1 + \frac{\alpha}{4dp_j}\right)\right] \leq \exp\left(-\frac{\alpha^2 p_j m}{48d^2(p_j)^2}\right) = \exp\left(-\frac{\alpha^2 m}{48d^2 p_j}\right) \leq \exp\left(-\frac{\alpha}{12d}\frac{1000d}{\alpha}\log\frac{dR}{\beta}\right) \leq \frac{\beta}{4d(R+1)},$$

where in the third inequality we used that $p_j \leq \frac{\alpha}{d}$ and substituted in a lower bound for $m$.

Secondly, when $p_j < \frac{\alpha}{4d}$, we use the multiplicative Chernoff bound for all $\gamma > 0$ from Claim F.2:

$$\Pr\left[\hat{p}_r[j] > p_j\left(1 + \frac{\alpha}{4dp_j}\right)\right] \leq \exp\left(-\frac{\alpha^2 p_j m}{16d^2 p_j^2(2 + \frac{\alpha}{4dp_j})}\right) \leq \exp\left(-\frac{\alpha^2 m}{12d^2 p_j(\frac{\alpha}{d})}\right)$$

$$= \exp\left(-\frac{\alpha m}{12dp_j}\right) \leq \exp\left(-\frac{\alpha}{12d}\frac{1000d\log(dR/\beta)}{\alpha}\right) \leq \frac{\beta}{4d(R+1)},$$

where in the first inequality we used that since $\frac{\alpha}{4dp_j} > 1$, $\frac{\alpha}{4dp_j} + 2 \leq 3\frac{\alpha}{4dp_j}$, and in the third inequality we substituted a lower bound for the value of $m$ and upper bounded $p_j$ by $\alpha$.

Similar inequalities hold for the lower tails of $\hat{p}_r[j]$. Taking a union bound over all $j \in [d]$ such that $p_j \leq \frac{\alpha}{d}$ completes the the proof of Item 1 in Lemma D.3.

Next, assume that $\frac{\alpha}{d} \leq p_j \leq \frac{1}{2}$. By the Chernoff bound from Claim F.2 for $\gamma \in (0,1)$,

$$\Pr\left[\hat{p}_r[j] - p_j \geq \frac{p_j}{16}\right] = \Pr\left[\hat{p}_r[j] \geq p_j\left(1 + \frac{1}{16}\right)\right] \leq \exp\left(-\frac{p_j m}{3 \cdot 16}\right) \leq \frac{\beta}{2d(R+1)},$$

where the final inequality holds since $p_j m \geq \frac{\alpha}{d} 1000 \frac{d}{\alpha} \log \frac{dR}{\beta}$. A similar bound holds for the lower tail of $\hat{p}_r[j]$. Taking a union bound over all $j \in [d]$, and all $r \in [2R+1]$ gives the result. $\qquad\square$

Next, we argue that truncation is unlikely in any round (given successful bucketing). Recall that $u_r = 1/2^r$ for all $r \in [R]$ (see Algorithm 4). For all $r \in [R]$, let $u_{R+r} = u_r$. Let $u_{2R+1} = 20/d$. A version of the following lemma is stated and proved in [19] (for us, the smallest upper bound of a bucket is $u_{2R+1} = 20/d$ instead of $1/d$, but the truncation ceiling $T_{2R+1}$ is also larger than in [19] to balance this out.)

**Lemma D.4** ([19], Claims 5.10 and 5.18). *Let $G_2$ be the following event that, for every round $r \in [2R+1]$, the following holds: if $p_j \leq u_r$ for all $j \in S_r$, then for every $i \in [m]$,*

$$\|x_i^r[S_r]\|_2 \leq T_r,$$

*that is, no rows are truncated in the calculation of $\text{tmean}_{T_r}(\mathbf{x}^r[S_r])$ in Steps 6 or 14 of Algorithm 4. Then, $\Pr\left[\overline{G_2}\right] \leq 3\beta$, where the probability is over the randomness of the input data and the coins of $\mathcal{A}_{prod}$.*

Finally, we prove that the amount of noise added in any round is unlikely to be large. For all $r \in [2R+1]$, let $Z_r$ be a $d$-dimensional random vector representing the noise added in round $r$ as in Steps 6 and 14 of Algorithm 4. For attributes $j \in [d]$ to which no noise is added in round $r$, the coordinate $Z_r[j] = 0$. The remaining $Z_r[j]$ are drawn from independent zero-mean Gaussians with standard deviation specified in Steps 6 and 14 of Algorithm 4.

**Lemma D.5.** *Let $G_3$ be the event that for all rounds $r \in [2R+1]$, for all $j \in S_r$,*

$$|Z_r[j]| \leq \frac{\alpha u_r}{100}.$$

*Then $\Pr\left[\overline{G_3}\right] \leq \beta$, where the probability is over the randomness of the input data and the coins of $\mathcal{A}_{prod}$.*

*Proof.* For rounds $r \in [2R]$, the standard deviation of univariate Gaussian noise $Z_r[j]$ added in round $r$ is $\sigma_r = \sqrt{\frac{3u_r|S_r|}{\rho m^2} \log \frac{mR}{\beta}}$. Set $t = \sqrt{2 \ln \frac{6dR}{\beta}}$. By Lemma F.3 on the concentration of a zero-mean Gaussian random variable along with a union bound,

$$\Pr(\max_{j \in S_r} |Z_r[j]| \geq t\sigma_r) \leq \sum_{j \in S_r} \Pr(|Z_r[j]| \geq t\sigma_r) \leq \sum_{j \in S_r} 2e^{-t^2/2} \leq \frac{\beta}{2R+1}.$$

Since $m \geq \frac{600d}{\alpha\sqrt{\rho}} \log^{5/4}(\frac{dR}{\alpha\beta\sqrt{\rho}})$ and , and because $u_r \geq \frac{40}{d}$, for all $r \in [2R]$,

$$t\sigma_r = \sqrt{\frac{6u_r|S_r| \log \frac{mR}{\beta} \ln \frac{6dR}{\beta}}{\rho m^2}} \leq \alpha\sqrt{\frac{6u_r|S_r| \log \frac{dR}{\alpha\beta\sqrt{\rho}} \ln \frac{6dR}{\beta}}{36000d^2 \log^{10/4}(\frac{dR}{\alpha\beta\sqrt{\rho}})}} \leq \alpha\sqrt{\frac{6u_r d}{3600d^2 \log^{1/4}(\frac{dR}{\beta})}} \leq \frac{\alpha u_r}{100},$$

where the first inequality is because $\log \frac{mR}{\beta}/m$ is a decreasing function for $m$ and hence we can upper bound the expression by using a lower bound of $m$. We also use the fact that the term $\log \frac{mR}{\beta} \leq 10 \log \frac{dR}{\alpha\beta\sqrt{\rho}}$). The second inequality follows by cancelling out some log terms and using the fact that $|S_r| \leq d$, and the last inequality follows because $\frac{1}{d} \leq \frac{u_r}{40}$, $\beta \leq 1$, and because $d$ is sufficiently large. For $r = 2R+1$, the standard deviation $\sigma_r = \sqrt{\frac{100}{\rho m^2} \log \frac{m}{\beta}}$, so with the same value of $t$, we get the same result. Taking a union bound over all $r \in [2R+1]$ gives the result. $\qquad\square$

The following corollary summarizes our analysis of good events and follows from Lemmas D.3–D.5 by a union bound.

**Corollary D.6.** *Let $G$ be the event $G_1 \cap G_2 \cap G_3$. Then, $\Pr[\overline{G}] \leq 6\beta$, where the probability is over the randomness of the input data and the coins of $\mathcal{A}_{prod}$.*

## D.3 Success of the Bucketing Phase

In this section, we argue that if the good event $G$ occurs, then the bucketing phase succeeds.

**Lemma D.7.** *Let $B$ be the event that that the bucketing procedure is successful, namely, for all rounds $r \in [R] \cup \{2R+1\}$ and for all coordinates $j \in [d]$, the following statements hold:*

1. *If $r \in [R]$ and $p_j \in S_{R+r}$, then $u_r/4 \leq p_j \leq u_r$.*

2. *If $p_j \in S_{2R+1}$ then $p_j \leq u_{2R+1}$.*

*If the good event $G$ defined in Corollary D.6 occurs then $B$ occurs.*

*Proof.* Assume that $B$ occurs. We prove this lemma by induction on $r$. Recall that $S_{R+r} = S_r \setminus S_{r+1}$ for all $r \in [R]$. To prove Item 1, we show that, for all rounds $r \in [R]$, if $j \in S_r$ then $p_j \leq u_r$, and if $j \notin S_r$ then $p_j \geq u_r/2$. For the first round (the base case of the induction), since $u_1 = 1/2$, and since by assumption $p_j \leq 1/2$ for all $j \in [d]$, we have that $p_j \leq u_1$. Additionally, since $S_1 = [d]$, it vacuously holds that $p_j \geq u_1/2$ for all $j \notin S_1$. Next, fix any $r \in [R-1]$. The inductive hypothesis is that for round $r$, if $j \in S_r$ then $p_j \leq u_r$ and if $j \notin S_r$ then $p_j \geq u_r/2 = u_{r+1}$.

We prove that this statement holds for round $r + 1$. For all $j \in S_r$, let $\tilde{p}_r[j]$ be the noisy empirical estimate obtained for coordinate $j$ in Step 6 of Algorithm 4 (in round $r$). By Item 1 of the definition of event $G_1$, for all $j \in S_r$ with $p_j > \frac{\alpha}{d}$,

$$|\hat{p}_r[j] - p_j| \leq \frac{p_j}{16} \leq \frac{u_r}{16},$$

where the second inequality is by the induction hypothesis. Similarly, by Item 2 of the definition of event $G_1$, for all $j \in S_r$ with $p_j \leq \frac{\alpha}{d}$,

$$|\hat{p}_r[j] - p_j| \leq \frac{\alpha}{4d} \leq \frac{u_r}{160},$$

where the second inequality holds since $u_r \geq \frac{40}{d}$ for all $r \in [R]$ and $\alpha \leq 1$.

By the inductive hypothesis, $p_j \leq u_r$ for all $j \in S_r$. Hence, by the definition of event $G_2$, no truncation occurs in round $r$. Also, $|Z_r[j]| \leq \frac{u_r}{100}$, by the definition of event $G_3$, since $\alpha \leq 1$. Hence, for all $j \in S_r$,

$$|\hat{p}_r[j] - \tilde{p}_r[j]| \leq \frac{u_r}{100}.$$

By the triangle inequality, we get that for all $j \in S_r$,

$$|\tilde{p}_r[j] - p_j| \leq |\hat{p}_r[j] - \tilde{p}_r[j]| + |\hat{p}_r[j] - p_j| \leq +\frac{u_r}{100} + \frac{u_r}{16} \leq \frac{u_r}{8}. \tag{16}$$

Fix any $j \in S_r$. Recall that $\tau_r = \frac{3u_{r+1}}{4}$. If $\tilde{p}_r[j] \leq \tau_r$ then, by (16), $p_j \leq \tau_r + \frac{u_r}{8} = \frac{3u_{r+1}}{4} + \frac{u_{r+1}}{4} = u_{r+1}$. Similarly, if $\tilde{p}_r[j] \geq \tau_r$, then $p_j \geq \frac{3u_{r+1}}{4} - \frac{u_{r+1}}{4} = \frac{u_{r+1}}{2}$.

This completes the inductive step and proves that at the beginning of round $r+1$, we have that $p_j \leq u_{r+1}$ for all $j \in S_{r+1}$, and $p_j \geq \frac{u_{r+1}}{2}$ for all $j \notin S_{r+1}$. Item 2 follows from an extension of the same argument. $\square$

## D.4 Success of the Sampling Phase

**Lemma D.8** (Success of sampling phase)**.** *For all $j \in [d]$, for $q[j]$ defined as in Step 16 of algorithm $\mathcal{A}_{prod}$, when $\mathcal{A}_{prod}$ is run with failure probability parameter $\beta \in (0, \frac{1}{12}]$ and target accuracy $\alpha \in (0, 1]$,*

1. *if $\frac{\alpha}{d} < p_j \le \frac{1}{2}$, then $|\mathbb{E}[q[j] - p_j]| \le 12\beta$;*

2. *if $p_j \le \frac{\alpha}{d}$, then $|\mathbb{E}[q[j] - p_j]| \le \dfrac{\alpha}{2d} + 6\beta$;*

*where the expectations are taken over the randomness of the data and the noise.*

*Proof.* We start by proving Item 1.

Fix any $j \in [d]$ with $\frac{\alpha}{d} \le p_j \le \frac{1}{2}$. First, we argue that if event $G$ occurs, then no noisy empirical means are clipped in the sampling phase. By construction, $(S_{R+1}, \ldots, S_{2R+1})$ is a partition of $[d]$. For all $j \in [d]$, let $r(j)$ denote the round $r \in \{R+1, \ldots, 2R+1\}$ such that $j \in S_r$. Now suppose $G$ occurred. By the triangle inequality and since $G$ implies $G_1, G_2, G_3$, and $B$,

$$|p_j - \tilde{p}_{r(j)}[j]| \le |p_j - \hat{p}_{r(j)}[j]| + |\hat{p}_{r(j)}[j] - \tilde{p}_{r(j)}[j]| \le \frac{p_j}{16} + |Z_{r(j)}[j]| \le \frac{p_j}{16} + \frac{\alpha u_{r(j)}}{100} \le \frac{p_j}{3}, \qquad (17)$$

where the second inequality is by the definition of event $G_1$, the fact that $G$ implies $B$, and the definition of event $G_2$ and the third inequality is by the definition of event $G_3$. The final inequality uses the fact that event $B$ occurs; if $p_j > \frac{5}{d}$ then $u_{r(j)} \le 4p_j$, and otherwise $u_{r(j)} = \frac{20}{d}$ and hence $\frac{\alpha u_{r(j)}}{100} = \frac{20\alpha}{100d} \le \frac{p_j}{5}$.

If $G$ occurs, by (17) and since $0 < p_j \le \frac{1}{2}$, we have

$$0 < \frac{2p_j}{3} \le \tilde{p}_{r(j)}[j] \le \frac{4p_j}{3} < 1,$$

and thus $\tilde{p}_{r(j)}[j]$ does not get clipped.

Next, by the law of total expectation,

$$\mathbb{E}[q[j] - p_j] = \mathbb{E}[q[j] - p_j \mid G] \cdot \Pr[G] + \mathbb{E}[q[j] - p_j \mid \overline{G}] \cdot \Pr[\overline{G}]$$

$$\le \mathbb{E}[q[j] \mid G] - p_j + 6\beta \le \frac{\mathbb{E}[\hat{p}_{r(j)}[j] + Z_{r(j)}[j]]}{\Pr[G]} - p_j + 6\beta$$

$$\le \frac{p_j}{1 - 6\beta} - p_j + 6\beta \le \frac{1}{2}\left(\frac{1}{1 - 6\beta} - 1\right) + 6\beta \le 12\beta,$$

where the first inequality holds by Corollary D.6 and the fact that $\mathbb{E}[q[j] - p_j] \le 1$. The second inequality uses the fact that when $G$ occurs, there is no clipping and truncation, and $\mathbb{E}[A \mid E] \le \mathbb{E}[A]/\Pr[E]$ for all random variables $A$ and events $E$. The third inequality is by the fact that $\mathbb{E}[\hat{p}_{r(j)}[j]] = p_j$ and $\mathbb{E}[Z_{r(j)}[j]] = 0$, and by Corollary D.6. The last inequality holds because $\beta \le \frac{1}{12}$ and $p_j \le \frac{1}{2}$ by assumption.

Analogously, $\mathbb{E}[p_j - q[j]] \le 12\beta$, which completes the proof of Item 1.

Next, we prove Item 2. Recall the event $G$ defined in Corollary D.6 and the event $B$ defined in Lemma D.7. Fix a coordinate $j \in [d]$ with $p_j \le \frac{\alpha}{d}$. By Lemma D.7, the law of total expectation, and the fact that $|\mathbb{E}[q[j] - p_j \mid \overline{B}]| \le 1$,

$$|\mathbb{E}[q[j] - p_j]| \le |\mathbb{E}[q[j] - p_j \mid G]| \cdot \Pr[G] + |\mathbb{E}[q[j] - p_j \mid \overline{B}]| \cdot \Pr[\overline{G}] \le |\mathbb{E}[q[j] - p_j \mid G]| + 6\beta. \qquad (18)$$

Now, we show that $|\mathbb{E}[q[j] - p_j \mid G] \le \frac{\alpha}{2d}$. Conditioned on event $G$, using Lemma D.7, event $B$ occurs, and the output bits for all coordinates $j$ with $p_j \le \frac{\alpha}{d}$ are sampled in round $2R + 1$. Conditioned on event $G$, using the fact that event $B$ occurs, and using the definition of event $G_2$ on truncation of empirical estimates,

$$|[\tilde{p}_{2R+1}[j]]_0^1 - p_j| \le |\tilde{p}_{2R+1}[j] - p_j| = |Z_{2R+1}[j]| \le \frac{\alpha u_{2R+1}}{100} \le \frac{\alpha}{2d},$$

where the second to last inequality is by the definition of event $G_3$, and the last inequality is since $u_{2R+1} = \frac{20}{d}$. Thus, $|\mathbb{E}[q[j] - p_j \mid G]| \le \frac{\alpha}{2d}$. Combining this with (18) proves Item 2 of Lemma D.8. $\qquad \square$

## D.5 Proof of Main Theorem

Finally, we use Lemma D.8 to prove the theorem.

*Proof of Theorem D.1.* Fix $\rho \in (0,1], \alpha \in (0,1), \beta = \frac{\alpha}{12d}$, and $R = \log_2(d/40)$. Fix the sample size

$$ n = \frac{1200(2R+1)d}{\alpha\sqrt{2\rho}} \log^{5/4} \frac{dR}{\alpha\beta\sqrt{2\rho}} = \tilde{O}\Big(\frac{d}{\alpha\sqrt{\rho}}\Big) $$

for this setting of $\beta$ and $R$.

First, by Lemma D.2, we have that $\mathcal{A}_{prod}$ is $\rho$-zCDP.

Next, we reason about accuracy. Let $Q_{\mathcal{A}_{prod}, P^{\otimes d}}$ be the distribution of the output of the sampler $\mathcal{A}_{prod}$ with randomness coming from the data and coins of the algorithm. Observe that $Q_{\mathcal{A}_{prod}, P^{\otimes d}}$ is a product distribution and that the marginal bias of each coordinate $j \in [d]$ is $\mathbb{E}[q[j]]$. Let the marginal distributions of $Q_{\mathcal{A}_{prod}, P^{\otimes d}}$ be $Q_1, \ldots, Q_d$. By the subadditivity of total variation distance between two product distributions,

$$
\begin{aligned}
d_{TV}(Q_{\mathcal{A}_{prod}, P}, P^{\otimes d}) &\leq \sum_{i=1}^{d} d_{TV}(Q_i, P) = \sum_{i=1}^{d} |\mathbb{E}[q[j] - p_j]| \\
&= \sum_{i:p_i > \frac{\alpha}{d}} |\mathbb{E}[q[j] - p_j]| + \sum_{i:p_i \leq \frac{\alpha}{d}} |\mathbb{E}[q[j] - p_j]| \\
&\leq \sum_{i:p_i > \frac{\alpha}{d}} 12\beta + \sum_{i:p_i \leq \frac{\alpha}{d}} \Big(\frac{\alpha}{2d} + 6\beta\Big) \leq \alpha,
\end{aligned}
$$

where we got the first equality by substituting the expression for the total variation distance between two Bernoulli distributions, the second inequality is by Lemma D.8 (since $\beta = \frac{\alpha}{12d} \leq \frac{1}{12}$, this lemma is applicable), and the final inequality holds because $\beta = \frac{\alpha}{12d}$. □

Finally, we complete this section by proving Theorem 1.6 from the introduction.

*Proof of Theorem 1.6.* Set $\rho = \frac{\epsilon^2}{16\log(1/\delta)}$. By Lemma A.4, for all $\delta \in (0, 1/2]$, algorithm $\mathcal{A}_{prod}$ is $(\epsilon, \delta)$-differentially private. Substituting this value of $\rho$ into Theorem D.1, we get that the sampler $\mathcal{A}_{prod}$ is $\alpha$-accurate for input datasets of size

$$ n = O\left(\frac{d}{\alpha\epsilon} \cdot \sqrt{\log\frac{1}{\delta}}\left(\log^{9/4} d + \log^{5/4}\frac{\log\frac{1}{\delta}}{\alpha\varepsilon}\right)\right). $$

For $\log(1/\delta) = polylog(n)$, we get that $\mathcal{A}_{prod}$ is $\alpha$-accurate for input datasets of size $n = \tilde{O}(\frac{d}{\alpha\epsilon})$. This proves the theorem, since $\delta \leq 1/2$. □

# E Products of Bernoulli Distributions with Bounded Bias

## E.1 Sampling Algorithms for Products of Bernoullis with Bounded Bias

In this section, we consider Bernoulli distributions and, more generally, products of Bernoulli distributions with bounded bias. We show that, when the bias is bounded, differentially private sampling can be performed with datasets of significantly smaller size than in the general case. For Bernoulli distributions with bounded bias, we achieve this (in Theorem E.1) with pure differential privacy, that is, with $\delta = 0$. For products of Bernoulli distributions, we give a zCDP algorithm (see Theorem E.4). Theorems E.1 and E.4, in conjunction with Lemma A.4 relating $\rho$-zCDP and differential privacy, directly yield Theorem 1.8. In Section E.2, we prove our lower bound for products of Bernoulli distributions with bounded bias, encapsulated in Theorem 1.9.

### E.1.1 Private Sampler for Bernoulli Distributions with Bounded Bias

First, we consider the class $\mathcal{B}_{[\frac{1}{3}, \frac{2}{3}]}$ of Bernoulli distributions (see Definition A.14) with an unknown bias $p \in \left[\frac{1}{3}, \frac{2}{3}\right]$. Even though class $\mathcal{B}_{[\frac{1}{3}, \frac{2}{3}]}$ is the hardest to learn privately among the classes of Bernoulli distributions, we show in the next theorem that private sampling from this class is easy.

**Theorem E.1.** *For all $\varepsilon > 0$ and $\alpha \in (0,1)$, there exists an $(\varepsilon, 0)$-differentially private sampler for the class $\mathcal{B}_{[\frac{1}{3}, \frac{2}{3}]}$ of Bernoulli distributions with bias in $\left[\frac{1}{3}, \frac{2}{3}\right]$ that is $\alpha$-accurate for datasets of size $n = O(\frac{1}{\varepsilon} + \ln \frac{1}{\alpha})$.*

*Proof.* We use $[\,\cdot\,]_a^b$ to denote rounding an arbitrary real number to the nearest value in $[a, b]$. Consider the following sampler $\mathcal{A}_{clip}$: on input $\mathbf{x} \in \{0,1\}^n$, compute the sample proportion $\hat{p} = \frac{1}{n} \sum_{i \in [n]} x_i$, obtain a clipped proportion $\tilde{p} = [\hat{p}]_{1/4}^{3/4}$, and output $b \sim \mathsf{Ber}(\tilde{p})$.

**Claim E.2.** *Sampler $\mathcal{A}_{clip}$ is $\alpha$-accurate on the class $\mathcal{B}_{[\frac{1}{3}, \frac{2}{3}]}$ with dataset of size $n \geq 72 \ln \frac{6}{\alpha}$.*

*Proof.* Define the "good" event $E$ that no rounding occurs when sample proportion is clipped, that is, $\tilde{p} = \hat{p}$. Since $p \in [1/3, 2/3]$,

$$\Pr[\overline{E}] = \Pr\left[\hat{p} \notin [1/4, 3/4]\right] \leq \Pr\left[|\hat{p} - p| \geq \frac{1}{12}\right] \leq 2e^{-n/72} \leq \frac{\alpha}{3}, \tag{19}$$

where we applied the Hoeffding bound (specifically, that $\Pr[|\hat{p} - \mathbb{E}[\hat{p}]| \geq t] \leq 2e^{-2nt^2}$) and our lower bound on $n$.

Let $Q$ be the distribution of the output bit $b$ for a dataset selected i.i.d. from $\mathsf{Ber}(p)$. Then, by Claim A.20 and the description of $\mathcal{A}_{clip}$,

$$d_{TV}(Q, \mathsf{Ber}(p)) = |\mathbb{E}(b) - p| = |\mathbb{E}(\tilde{p}) - \mathbb{E}(\hat{p})|.$$

Next, we observe that $\mathbb{E}[\tilde{p}|E] = \mathbb{E}[\hat{p}|E]$ and use (19) to bound $d_{TV}(Q, \mathsf{Ber}(p))$. Specifically,

$$\mathbb{E}[\hat{p}] = \mathbb{E}[\hat{p}|E] \cdot \Pr[E] + \mathbb{E}[\hat{p}|\overline{E}] \cdot \Pr[\overline{E}] \leq \mathbb{E}[\hat{p}|E] \cdot 1 + 1 \cdot \Pr[\overline{E}]$$

$$\leq \mathbb{E}[\hat{p}|E] + \frac{\alpha}{3} = \mathbb{E}[\tilde{p}|E] + \frac{\alpha}{3} \leq \frac{\mathbb{E}[\tilde{p}]}{\Pr[E]} + \frac{\alpha}{3} \leq \frac{\mathbb{E}[\tilde{p}]}{1 - \alpha/3} + \frac{\alpha}{3}$$

$$\leq \mathbb{E}[\tilde{p}] + \left(\frac{1}{1 - \alpha/3} - 1\right) + \frac{\alpha}{3} \leq \mathbb{E}[\tilde{p}] + \frac{\alpha}{2} + \frac{\alpha}{3} \leq \mathbb{E}[\tilde{p}] + \alpha.$$

Similarly, $\mathbb{E}[\tilde{p}] \leq \mathbb{E}[\hat{p}] + \alpha$. We get that $d_{TV}(Q, \mathsf{Ber}(p)) = |\mathbb{E}(\tilde{p}) - \mathbb{E}(\hat{p})| \leq \alpha$, completing the accuracy analysis. $\qquad\square$

**Claim E.3.** *Sampler $\mathcal{A}_{clip}$ is $(4/n, 0)$-differentially private.*

*Proof.* By definition of the sampler, $\Pr[b = 1] = \tilde{p}$. Consider two datasets $\mathbf{x}$ and $\mathbf{x}'$ that differ in one record. The sample proportions $\hat{p} = \frac{1}{n} \sum_{i \in [n]} x_i$ and $\hat{p}' = \frac{1}{n} \sum_{i \in [n]} x_i'$ differ by at most $1/n$. Let $\tilde{p}$ and $\tilde{p}'$ be the corresponding clipped proportions, which also differ by at most $1/n$. Then, since $\tilde{p} \geq 1/4$,

$$\tilde{p}' \leq \tilde{p} + \frac{1}{n} \leq \tilde{p} + \tilde{p}\frac{4}{n} = \tilde{p}\left(1 + \frac{4}{n}\right) = \tilde{p} \cdot e^{4/n},$$

where we used the fact that $1 + t \leq e^t$ for all $t$. Similarly, since $\tilde{p} \leq 3/4$, we get that the probabilities of returning $b = 0$ for inputs $\mathbf{x}$ and $\mathbf{x}'$ also differ by at most a factor of $e^{4/n}$. Thus, $\mathcal{A}_{clip}$ is $4/n$-differentially private. $\qquad\square$

Now we set $n \geq \max\left\{72 \ln \frac{6}{\alpha}, \frac{4}{\varepsilon}\right\}$ and use Claims E.2 and E.3 to get both accuracy and privacy guarantees. Observe that when $n \geq 4/\varepsilon$, we get that $4/n \leq \varepsilon$, that is, $\mathcal{A}_{clip}$ is $(\varepsilon, 0)$-differentially private. This completes the proof of Theorem E.1. $\qquad\square$

### E.1.2 Private Sampler for Products of Bernoulli Distributions with Bounded Bias

In this section, we consider product distributions, where each marginal is a Bernouli distribution with a bias between $1/3$ and $2/3$. For this class, significantly fewer samples are needed for private sampling than for the general class of products of Bernoulli distributions.

**Theorem E.4.** *For all all $\rho > 0$ and $\alpha \in (0,1)$, there exists a $\rho$-zCDP sampler for the class $\mathcal{B}_{[\frac{1}{3},\frac{2}{3}]}^{\otimes d}$ of products of Bernoulli distributions with bias in $\left[\frac{1}{3}, \frac{2}{3}\right]$ that is $\alpha$-accurate for datasets of size $n = O\left(\frac{\sqrt{d}}{\sqrt{\rho}} + \log \frac{d}{\alpha}\right)$.*

*Proof.* The input to a sampler for $\mathcal{B}_{[\frac{1}{3},\frac{2}{3}]}^{\otimes d}$ is an $n \times d$ matrix $\mathbf{x} \in \{0,1\}^{n \times d}$, where row $i$ contains the $i$-th record and column $j$ contains all input bits for $j$-th attribute. Recall sampler $\mathcal{A}_{clip}$ from the proof of Theorem E.1. For each $j \in [d]$, our sampler runs sampler $\mathcal{A}_{clip}$ on column $j$ of $\mathbf{x}$ and records its output bit $b_j$; it returns the vector $b = (b_1, \ldots, b_d)$.

We show that this sampler is $\alpha$-accurate on the class $\mathcal{B}_{[\frac{1}{3},\frac{2}{3}]}^{\otimes d}$ with $n \geq 72 \ln \frac{6d}{\alpha}$ samples. Let $P = P_1 \otimes \cdots \otimes P_d$ be the input product distribution, where $P_j = \mathsf{Ber}(p_j)$ for all coordinates $j \in [d]$. Let $Q = Q_1 \otimes \cdots \otimes Q_d$ be the distribution of the output vector $b$ for a dataset selected i.i.d. from $P$. (Since the coordinates of $b$ are mutually independent, $Q$ is indeed a product distribution.) By Claim E.2 applied with $\alpha/d$ as accuracy parameter, $d_{TV}(Q_j, P_j) \leq \frac{\alpha}{d}$. By subadditivity of the statistical distance between two product distributions,

$$d_{TV}(Q, P) \leq \sum_{j \in [d]} d_{TV}(Q_j, P_j) \leq d \cdot \frac{\alpha}{d} = \alpha,$$

completing the proof that our sampler is $\alpha$-accurate.

Finally, we show that our sampleris $\rho$-zCDP when $n \geq \sqrt{8d/\rho}$. The sampler is a composition of $d$ algorithms, each returning one bit. By Claim E.3, these algorithms are $(4/n, 0)$-differentially private and, consequently, also $\frac{8}{n^2}$-zCDP. A composition of $d$ such algorithms is then $\frac{8d}{n^2}$-zCDP. That is, when $n \geq \sqrt{8d/\rho}$, it is $\rho$-zCDP, as required. □

## E.2 Lower Bound for Products of Bernoullis with Bounded Bias

We now prove a lower bound that matches the guarantees of the algorithm of the previous section.

**Theorem E.5** (Theorem 1.9, restated). *For all sufficiently small $\alpha > 0$, and for all $d, n \in \mathbb{N}$, $\varepsilon \in (0,1]$, and $\delta \leq \frac{1}{100n}$, if there exists an $(\varepsilon, \delta)$-differentially private sampler that is $\alpha$-accurate on the class of products of $d$ Bernoulli distributions with biases in $\left[\frac{1}{3}, \frac{2}{3}\right]$ on datasets of size $n$, then $n = \Omega(\sqrt{d}/\varepsilon)$.*

To prove the theorem, we reduce the problem of accurately estimating the marginal biases of a product distribution over $\{0,1\}^d$ to the problem of sampling from the product distribution. This involves dividing the dataset into a constant number of disjoint parts and passing each part separately to a sampler for product distributions to obtain a constant number of independent samples. Then, by averaging the samples obtained, we get an estimate of the marginal biases. We also observe that a marginal estimator for the class $\mathcal{B}_{[\frac{1}{3},\frac{2}{3}]}^{\otimes d}$ can be converted into a marginal estimator for the class $\mathcal{B}^{\otimes d}$ with only a constant factor loss in accuracy. To do this, we flip every bit of every sample with probability $1/3$, which gives us a dataset that looks like it is drawn from a product distribution in $\mathcal{B}_{[\frac{1}{3},\frac{2}{3}]}^{\otimes d}$. We then use the marginal estimator for $\mathcal{B}_{[\frac{1}{3},\frac{2}{3}]}^{\otimes d}$ and transform the estimated biases back to the original range by multiplying by 3 and subtracting 1. Finally, applying the lower bound of Bun et al. [11] for the sample complexity of marginal estimation for the class $\mathcal{B}^{\otimes d}$, we obtain a lower bound on the sample complexity for the problem of accurately sampling from product distributions with bounded biases.

**Definition E.6** (Marginal Estimator). *For $\alpha', \beta', \gamma \in [0,1]$, and a class $\mathcal{C}$ of distributions on $\{0,1\}^d$, an algorithm $\mathcal{M}$ is an $(\alpha', \beta', \gamma, \mathcal{C})$-marginal estimator with sample size $n$ if, given $\mathbf{X} \sim P^{\otimes n}$ where $P \in \mathcal{C}$, with probability at least $1 - \gamma$, algorithm $\mathcal{M}$ returns $\tilde{p}_1, \ldots, \tilde{p}_d$ such that*

$$|\{j \in [d] : |p_j - \tilde{p}_j| > \alpha'\}| < \beta' d.$$

---

**Algorithm 5** Marginal Estimator $\mathcal{M}_c$ for $\mathcal{B}_{[\frac{1}{3},\frac{2}{3}]}^{\otimes d}$

---

    **Input:** dataset $\mathbf{x} \in \{0,1\}^{cn \times d}$, constant $c$, query access to sampler $\mathcal{A}$
    **Output:** marginal estimates $\tilde{\mathbf{p}} = (\tilde{p}_1, \ldots, \tilde{p}_d)$
1: Partition dataset $\mathbf{x}$ into $c$ equal parts: $\mathbf{x}^{(1)}, \ldots, \mathbf{x}^{(c)}$
2: **for** $i = 1$ to $c$ **do**:
3:     $Y_i \leftarrow \mathcal{A}(\mathbf{x}^{(i)})$                               ▷ Get $c$ independent samples from $\mathcal{A}$
4: $\tilde{\mathbf{p}} \leftarrow \frac{1}{c} \sum_{i=1}^{c} Y_i$                           ▷ Compute marginal estimates
5: **return** $\tilde{\mathbf{p}}$

---

**Lemma E.7** (Reduction from Marginal Estimation to Sampling). *For all $\alpha, \beta_0, \gamma_0 \in (0,1)$, there exists $c \in \mathbb{N}$ such that for all $\varepsilon, \delta > 0$: if $\mathcal{A}$ is an $(\varepsilon, \delta)$-DP sampler that is $\alpha$-accurate on class $\mathcal{B}_{[\frac{1}{3},\frac{2}{3}]}^{\otimes d}$ with sample size $n$, then $\mathcal{M}_c$ (Algorithm 5) is an $(\varepsilon, \delta)$-DP, $(2\alpha, \beta_0, \gamma_0, \mathcal{B}_{[\frac{1}{3},\frac{2}{3}]}^{\otimes d})$-marginal estimator with sample size $cn$.*

*Proof.* Fix a distribution $P \in \mathcal{B}_{[\frac{1}{3},\frac{2}{3}]}^{\otimes d}$. Let $p$ represent the vector of biases corresponding to $P$, and $Y_i$ for $i \in [c]$ and $\tilde{p}$ be as defined in Steps 3 and 4 of Algorithm 5. Consider any index $j \in [d]$. The expectations $\mathbb{E}[Y_i[j]]$ are the same for all $i \in [c]$. Define $q[j] = \mathbb{E}[Y_1[j]]$. Since $\mathcal{A}$ is $\alpha$-accurate with sample size $n$, we have $|q[j] - p_j| \leq \alpha$. Let $D$ be a positive constant to be set later. By Hoeffding's inequality (Claim F.4), $\Pr(|q[j] - \tilde{p}_j| \geq \frac{D}{\sqrt{c}}) \leq 2e^{-2D^2}$. By the triangle inequality, with probability at least $1 - 2e^{-2D^2}$,

$$|\tilde{p}_j - p_j| \leq |\tilde{p}_j - q[j]| + |q[j] - p_j| \leq \alpha + \frac{D}{\sqrt{c}}. \tag{20}$$

Since (20) holds for all $j \in [d]$, the expected number of $j \in [d]$ such that $|q[j] - p_j| > \alpha + \frac{D}{\sqrt{c}}$ is at most $2de^{-2D^2}$. By Markov's inequality,

$$\Pr(|\{j \in [d] : |p_j - \tilde{p}_j| > \alpha + \tfrac{D}{\sqrt{c}}\}| \geq \tfrac{2d}{\gamma_0} e^{-2D^2})) \leq \gamma_0.$$

Setting $D^2 = \frac{1}{2} \ln(\frac{2}{\beta_0 \gamma_0})$ ensures $\frac{2d}{\gamma_0} e^{-2D^2} \leq \beta_0 d$. Setting $c = \left\lceil \frac{D^2}{\alpha^2} \right\rceil$ further ensures that $\alpha + \frac{D}{\sqrt{c}} \leq 2\alpha$. We thus get the desired accuracy guarantee on $\mathcal{M}_c$ when $c = \left\lceil \frac{1}{2\alpha^2} \ln(\frac{2}{\beta_0 \gamma_0}) \right\rceil$.

Finally, changing one entry in the dataset $\mathbf{x}$ changes a single entry in only one of the parts $\mathbf{x}^{(i)}$, and only this part is fed to the $i^{th}$ call to $\mathcal{A}$. Since $\mathcal{A}$ is $(\varepsilon, \delta)$-DP, so is $\mathcal{M}_c$. This proves the lemma. $\qquad\square$

Next we show how to transform a marginal estimator for a product of bounded Bernoulli distributions into a marginal estimator for a product of arbitrary Bernoulli distributions. Let $BSC_{1/3}$ denote a *binary symmetric channel* with bias 1/3. That is, on input $\mathbf{x} \in \{0,1\}^{n \times d}$, each bit gets flipped independently with probability 1/3. In particular, $BSC_{1/3}(\mathbf{x}) = \mathbf{x} \oplus \mathbf{Z}$, where $\mathbf{Z} \sim \mathsf{Ber}(1/3)^{\otimes n \times d}$.

---

**Algorithm 6** Marginal Estimator $\mathcal{M}_{\mathsf{Ber}}$ for $\mathcal{B}^{\otimes d}$

---

    **Input:** dataset $\mathbf{x} \in \{0,1\}^{n \times d}$, query access to marginal estimator $\mathcal{M}$ for $\mathcal{B}_{[\frac{1}{3},\frac{2}{3}]}^{\otimes d}$
    **Output:** marginal estimates $\tilde{\mathbf{p}} = (\tilde{p}_1, \ldots, \tilde{p}_d)$
1: $\mathbf{x}^* \leftarrow BSC_{1/3}(\mathbf{x})$                            ▷ Change initial distribution
2: $\mathbf{p}^* \leftarrow \mathcal{M}(\mathbf{x}^*)$                              ▷ Get empirical estimates
3: $\tilde{\mathbf{p}} \leftarrow (3 \cdot p_1^* - 1, \ldots, 3 \cdot p_d^* - 1)$            ▷ Rescale empirical estimates
4: **return** $\tilde{\mathbf{p}}$

---

**Lemma E.8** (Reduction from General to Bounded Biases). *If $\mathcal{M}$ is an $(\alpha', \beta', \gamma, \mathcal{B}_{[\frac{1}{3},\frac{2}{3}]}^{\otimes d})$-marginal estimator with sample size $n$, then $\mathcal{M}_{\mathsf{Ber}}$ (in Algorithm 6) is a $(3\alpha', \beta', \gamma, \mathcal{B}^{\otimes d})$-marginal estimator with sample size $n$. If $\mathcal{M}$ is $(\varepsilon, \delta)$-differentially private, then so is $\mathcal{M}_{\mathsf{Ber}}$.*

*Proof.* We begin with the accuracy proof. Fix an $(\alpha', \beta', \gamma, \mathcal{B}_{[\frac{1}{3}, \frac{2}{3}]}^{\otimes d})$-marginal estimator $\mathcal{M}$ with sample size $n$. Fix a distribution $P \in \mathcal{B}^{\otimes d}$ with biases $\mathbf{p} = (p_1, \ldots, p_d)$. Let $\mathbf{X} \sim P^{\otimes n}$. Denote the output distribution of $BSC_{1/3}(\mathbf{X})$ by $P'$ and its biases by $\mathbf{p}' = (p'_1, \ldots, p'_d)$. Let $BSC_{1/3}(\mathbf{X})_i^j = \mathbf{X}_i^j \oplus \mathbf{Z}$ be the output for the $j$th attribute on the $i$th data record, where $Z_i^j \sim \mathsf{Ber}(1/3)$. Then, for all $j \in [d]$ and all $i \in [n]$,

$$p'_j = \Pr[BSC_{1/3}(\mathbf{X})_i^j = 1] = \Pr[X_i^j \oplus Z_i^j = 1] = \Pr[X_i^j = 1 \land Z_i^j = 0] + \Pr[X_i^j = 0 \land Z_i^j = 1]$$
$$= p_j \cdot \frac{2}{3} + (1 - p_j) \cdot \frac{1}{3} = \frac{p_j}{3} + \frac{1}{3}.$$

Thus, $BSC_{1/3}(\mathbf{X}) \in \mathcal{B}_{[\frac{1}{3}, \frac{2}{3}]}^{\otimes d}$, as desired. Since $\mathcal{M}$ is a $(\alpha', \beta', \gamma, \mathcal{B}_{[\frac{1}{3}, \frac{2}{3}]}^{\otimes d})$-marginal estimator with sample size $n$, estimator $\mathcal{M}$ returns $(p_1^*, \ldots, p_d^*)$ such that with probability at least $1 - \gamma$,

$$|\{j \in [d] : |p'_j - p_j^*| > \alpha'\}| < \beta' d. \tag{21}$$

Substituting $p'_j = p_j/3 + 1/3$ in the left-hand side of in (21) and then using $\tilde{p}_j = 3p_j^* + 1$, we get

$$|\{j \in [d] : |\frac{p_j}{3} + \frac{1}{3} - p_j^*| > \alpha'\}| = |\{j \in [d] : |p_j - (3p_j^* - 1)| > 3\alpha'\}|$$
$$= |\{j \in [d] : |p_j - \tilde{p}_j| \le 3\alpha'\}|.$$

Thus, $\mathcal{M}_{\mathsf{Ber}}$ is a $(3\alpha', \beta', \gamma, \mathcal{B}^{\otimes d})$-marginal estimator with sample size $n$.

Finally, we show that $\mathcal{M}_{\mathsf{Ber}}$ is differentially private. Suppose $\mathcal{M}$ is $(\varepsilon, \delta)$-differentially private. Fix neighboring datasets $\mathbf{x}$ and $\mathbf{x}'$ that differ in record $i$. Then, $BSC_{1/3}(\mathbf{x})$ and $BSC_{1/3}(\mathbf{x}')$ still only differ on record $i$. Since the output of $\mathcal{M}_{\mathsf{Ber}}$ is a post-processing of $\mathcal{M}$, we have that $\mathcal{M}_{\mathsf{Ber}}$ is $(\varepsilon, \delta)$-differentially private. □

We prove our main result by combining Lemmas E.7–E.8 with a lower bound on marginal estimation that is obtained using the fingerprinting codes technique of Bun, Ullman and Vadhan [11]. We use a corollary of the version of the result from [11] presented by Kamath and Ullman [20].

**Theorem E.9** (Consequence of [20], Theorem 3.3)**.** *Suppose there exists a function $n = n(d)$, such that for every $d \in \mathbb{N}$, there is a $(\alpha_0, \beta_0, \gamma_0, \mathcal{B}^{\otimes d})$-marginal estimator $\mathcal{M} : \{0, 1\}^{n \times d} \to \mathbb{R}^d$ that is $(\varepsilon, \frac{1}{100n})$-DP, where $\alpha_0, \beta_0, \gamma_0 \in (0, 1)$ are sufficiently small absolute constants. Then $n = \Omega(\sqrt{d}/\varepsilon)$.*

*Proof of Theorem E.5.* Let $\varepsilon, \delta \in (0, 1]$ with $\delta < \frac{1}{100n}$. Let $\alpha_0, \beta_0, \gamma_0$ be the constants from Theorem E.9, and set $\alpha = \frac{\alpha_0}{6}$. Let $\mathcal{A}$ be an $\alpha$-accurate, $(\varepsilon, \delta)$-DP sampler for the class $\mathcal{B}_{[\frac{1}{3}, \frac{2}{3}]}^{\otimes d}$ for datasets of some size $n$. By Lemma E.7, there exists an $(\varepsilon, \delta)$-differentially private, $(\frac{\alpha_0}{3}, \beta_0, \gamma_0, \mathcal{B}_{[\frac{1}{3}, \frac{2}{3}]}^{\otimes d})$-marginal estimator $\mathcal{M}_c$ for datasets of size $cn$ for an absolute constant $c = c(\alpha, \beta_0, \gamma_0)$. By Lemma E.8, $\mathcal{M}_{\mathsf{Ber}}$ defined in Algorithm 6 is a $(\varepsilon, \delta)$-differentially private, $(\alpha_0, \beta_0, \gamma_0, \mathcal{B}^{\otimes d})$-marginal estimator for datasets of size $cn$. By Theorem E.9, $n = \Omega(\sqrt{d}/\varepsilon)$, as desired.

□

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

# F  Inequalities Used in Technical Sections

We argue that the moments of the average of several identically distributed random variables are no larger than the corresponding moments of the individual random variables.

**Claim F.1.** *If random variables $A_1, \ldots, A_k$ are identically distributed, then, for all $\lambda > 0$,*

$$\mathbb{E}\left[\left(\frac{1}{k}\sum_{i=1}^{k} A_i\right)^{\lambda}\right] \leq \mathbb{E}\left[A_1^{\lambda}\right].$$

*Proof.* By Jensen's inequality, $\left(\frac{1}{k}\sum_{i=1}^{k} A_i\right)^{\lambda} \leq \frac{1}{k}\sum_{i=1}^{k} A_i^{\lambda}$ for any fixed values of $A_1, \ldots, A_k$. We take expectation on both sides, then use the linearity of expectation and that $A_1, \ldots, A_k$ are identically distributed:

$$\mathbb{E}\left[\left(\frac{1}{k}\sum_{i=1}^{k} A_i\right)^{\lambda}\right] \leq \mathbb{E}\left[\frac{1}{k}\sum_{i=1}^{k} A_i^{\lambda}\right] = \frac{1}{k}\sum_{i=1}^{k} \mathbb{E}\left[A_i^{\lambda}\right] = \mathbb{E}\left[A_1^{\lambda}\right].$$

$\square$

## F.1 Concentration Inequalities

**Claim F.2** (Chernoff Bounds). *Let $A$ be the average of $m$ independent 0-1 random variables with $\mu = \mathbb{E}[A]$. For $\gamma \in (0, 1)$,*

$$\Pr[A \geq \mu(1+\gamma)] \leq e^{-\frac{\gamma^2 \mu m}{3}};$$
$$\Pr[A \leq \mu(1-\gamma)] \leq e^{-\frac{\gamma^2 \mu m}{2}}.$$

*For $\gamma \geq 0$,*

$$\Pr[A \geq \mu(1+\gamma)] \leq e^{-\frac{\gamma^2 \mu m}{2+\gamma}};$$
$$\Pr[A \leq \mu(1-\gamma)] \leq e^{-\frac{\gamma^2 \mu m}{2+\gamma}}.$$

**Claim F.3** ([19], Lemma 2.8, Gaussian Concentration). *If $A$ is drawn from $\mathcal{N}(0, \sigma^2)$, then, for all $t > 0$,*

$$\Pr\left(|A| > t\sigma\right) \leq 2e^{-t^2/2}.$$

**Claim F.4** (Hoeffding's Inequality). *Let $A$ be the average of $m$ independent random variables in the interval $[0, 1]$ with $\mu = \mathbb{E}[A]$. For $h \geq 0$,*

$$\Pr[A - \mu \geq h] \leq e^{-2mh^2}.$$
$$\Pr[\mu - A \geq h] \leq e^{-2mh^2}.$$

# G   Lemmas on Amplification by Subsampling

**Definition G.1** ([22], Definition 3). *An algorithm $\mathcal{A}$ is $(\beta, \varepsilon, \delta)$-DPS if and only if $\beta > \delta$ and the algorithm $\mathcal{A}^{\beta}$ is $(\varepsilon, \delta)$-DP where $\mathcal{A}^{\beta}$ denotes the algorithm to first sample with probability $\beta$ (include each tuple in the input dataset with probability $\beta$), and then apply $\mathcal{A}$ to the sampled dataset.*

**Theorem G.2** ([22], Theorem 1). *Any $(\beta_1, \varepsilon_1, \delta_1)$-DPS algorithm is also $(\beta_2, \varepsilon_2, \delta_2)$-DPS for any $\beta_2 < \beta_1$ where $\varepsilon_2 = \ln\left(1 + \left(\frac{\beta_2}{\beta_1}(e^{\varepsilon_1} - 1)\right)\right)$, and $\delta_2 = \frac{\beta_2}{\beta_1}\delta_1$.*