# OpenReview forum: "Differentially Private Sampling from Distributions"
_NeurIPS.cc/2021/Conference — NeurIPS 2021 Poster_

### Official Review · Reviewer_behk · 2021-07-12

**Rating:** 6
**Confidence:** 3

**Summary:**

The paper studies the problem of sampling from a distribution in a differentially private manner.  More formally, the learner is given i.i.d samples from some unknown distribution $P$ over a domain $\mathcal{U}$ and its goal is to output an element of $\mathcal{U}$ such that the distribution of the output is close to $P$ in TV distance.  The problem of sampling is easier than the problem of learning the actual distribution $P$ and the authors show that in some cases it can be done using significantly fewer samples.


The main results are that for arbitrary discrete distributions over $k$, the sample complexity of differentially private sampling is $\theta(k/(\alpha \epsilon))$ (compared to $k/\alpha^2$ for learning and $k/\alpha^2 + k/(\alpha \epsilon)$ for private learning where $\alpha$ is the desired accuracy and $\epsilon$ is the privacy parameter).  For product distributions on $\{ 0,1 \}^d$, the main result is an upper bound of $\widetilde{O}(d/(\alpha \epsilon))$ although this can be improved to $\sqrt{d}/\epsilon + \log (1/\alpha)$ if the coordinates have means bounded away from $0$ and $1$.


**Main Review:**

Overall, the contributions are solid and the concept of sampling with differential privacy seems nice and novel.  The presentation is reasonable, although I found the choice to focus on the lower bound for arbitrary discrete distributions in the main paper (deferring the rest to the supplement) a bit puzzling.  In think the results for product distributions are more interesting while the results for arbitrary discrete distributions are mostly unsurprising/straight-forward.  For product distributions it is a bit unsatisfying that there  is a  simple algorithm for the case when the means are bounded away from $0$ and $1$ but the general case involves much more work and a much more complicated algorithm with worse guarantees.


After Author Response:  Thanks for the clarifications!  My score remains unchanged and I still think that the paper is a solid, but not quite spectacular contribution.

**Time Spent Reviewing:**

2

---

> ### Author Response · Authors · 2021-08-10
> **Response to Review**
>
> Thank you for the thoughtful review.
>
> The reviewer writes: _“I found the choice to focus on the lower bound for arbitrary discrete distributions in the main paper (deferring the rest to the supplement) a bit puzzling.”_ Unfortunately, the space limitation made it hard to include more than one nontrivial argument. We chose the lower bound for arbitrary discrete distributions because we found it to be the more conceptually interesting of the lower bounds. Because the existing information-theoretic frameworks used in lower bounds for learning do not apply to sampling, we gave a novel, direct argument which actually shows that generating an element from the support of $P$ is hard. In particular, the instances arising in the existing learning lower bounds (in which all elements have large probability) are easy for sampling and the instances arising in our lower bounds are easy for learning.
>
> Additionally, we wish to address the following comment: _“For product distributions it is a bit unsatisfying that there is a simple algorithm for the case when the means are bounded away from 0 and 1 but the general case involves much more work and a much more complicated algorithm with worse guarantees.”_
>
> The reviewer raises two issues with our algorithm for general product distributions: it is complicated and has high sample complexity. The latter may well be unavoidable. For example, with a single Bernoulli ($d=1$), the bounded-bias setting allows to (nearly) eliminate the dependency on $\alpha$, whereas the sample complexity must scale proportionally to $1/\alpha$ in the general case. We conjecture that a sample complexity of $\Omega(d/\alpha\epsilon)$ is necessary in the general, $d$-dimensional setting.
>
> As for the algorithm’s simplicity: we were not able to find a significantly simpler algorithm, despite considerable effort. We are working on simplifying the write-up and streamlining the analysis to make the algorithm more readable. The complicated part is the first phase of the algorithm (based closely on [KLSU]), which uses half the data to partition the coordinates according to a rough estimate of their bias. We don’t know how to get away without some iterative procedure along those lines.

---

### Official Review · Reviewer_b21p · 2021-07-14

**Rating:** 5
**Confidence:** 4

**Summary:**

The paper considers the following question: say you observe n data points, drawn iid from some distribution p, and you know that p belongs to some class of distributions P, then how large does n have to be such that you can use the data to generate a single new point that is close in total variation distance to the underlying distribution p, while being differentially private? In the paper, this question is referred to as the sample complexity of differentially private sampling.

The paper then investigates this question for two classes of distributions: any distribution supported on k points, and product distributions on {0,1}^d. It shows that for all the settings, the sample complexity of differentially private sampling is a lower bound for the sample complexity of differentially private learning, and that in some settings, the sample complexity for sampling is strictly smaller than that of learning.


**Limitations And Societal Impact:**

Yes

**Main Review:**

Essential points:

My main criticism of the paper is with the problem formulation. The paper currently motivates its set up with the question of sampling from a target distribution p using iid data drawn from p. But this setup does not really capture the real difficulty of sampling, which usually lies in the amount of computation needed to generate a sample, or the number of queries to the target distribution needed. As a result, the paper’s setup results in the task of trying to find a distribution that is close in total variation to the target distribution, which is really quite similar to the task of learning distributions.

The only difference, between this paper’s setup and the learning setup, is that in the objective of this paper, the expectation over given data is taken inside the total variation distance, whereas in the learning setup, the expectation is taken outside the total variation distance. This difference seems a bit artificial to me. The paper also did not provide any concrete motivating questions that suggest their framework is worth studying, which makes me unsure whether researchers who are not specialists of differential privacy will be interested in the setting of the paper.

Moreover, once one realizes that taking the expectation inside the total variation distance is the main difference between the paper’s setup and and the learning setup, some of the paper’s results then seem fairly natural. For instance, the paper notes that in the Bernoulli setting, the hard instances for learning are when the bias is close to ½, whereas these instances are easy for sampling. But that is not surprising, because if we take the expectation inside the total variation distance, then we don’t need to determine the exact bias.

The contribution of this paper seems to be mostly a technical one. Although the technical work is good, the arguments of this paper seem quite specific, and it is unclear if the general audience will be interested in the details.


**Time Spent Reviewing:**

2

---

> ### Author Response · Authors · 2021-08-10
> **Response to Review**
>
> Thank you for the thoughtful review.
>
> The reviewer writes: _“the paper’s setup results in the task of trying to find a distribution that is close in total variation to the target distribution […] The only difference between this paper’s setup and the learning setup, is that in the objective of this paper, the expectation over given data is taken inside the total variation distance [...]”_
>
> We find that this summary doesn’t do justice to the problem we consider. The difference between learning and sampling is not only in how the algorithm is evaluated, but also in the type of the output required. A sampling algorithm is not asked to output a description of a distribution, but rather a single draw from a distribution (that is close to the input distribution $P$). For example, for the class of Bernoulli distributions, a learning algorithm would output a bias (a number between 0 and 1) whereas a sampling algorithm has to output one coin flip (that is, 0 or 1). Nonprivately, the task of sampling is trivial: an algorithm that just outputs its first input record perfectly samples from any input distribution P; in contrast, non-private learning is far from trivial. Fundamentally, sampling and learning are different problems.
>
> One can view the output of a sampling algorithm as a degenerate distribution over a single element (the one that was output). For example, in the case of Bernoulli distributions, it would be either a distribution that places all the weight on 0 or a distribution that places all the weight on 1. In that case, the requirement is indeed that the expectation of this distribution be close to $P$. But this view doesn’t really get at what makes the problem interesting, since it does not capture the requirement of outputting a data element rather than a distribution. In particular, several of our lower bounds actually show that the task of _privately generating any element from the support of $P$_ requires many samples.
>
> Private learning and private sampling turn out to be very different problems, as evidenced by the fact that techniques used for analyzing learning algorithms do not work for analyzing sampling algorithms.
>
> The reviewer writes: _“The paper also did not provide any concrete motivating questions that suggest their framework is worth studying, which makes me unsure whether researchers who are not specialists of differential privacy will be interested in the setting of the paper.”_
>
> Our formulation of sampling captures settings where one wants to output data that “looks like” the population from which the dataset was drawn. For example, one might need realistic data for debugging a software program or for getting a quick idea of the range of values in a dataset. More broadly, our conceptual goal was to understand tasks that are weaker than learning, but still meaningful. Lower bounds for weaker tasks imply lower bounds for stronger ones; they also shed light on what aspects of the stronger problem really make it challenging. For example, to the extent that private distribution learning is harder than non-private distribution learning, this difficulty is completely explained (for some classes of distributions and some parameter settings) by the difficulty of private sampling. We will add more discussion and motivation in future versions of the paper (expanding on what we have in lines 25-28 and 132-134).
>
> Finally, we were puzzled by the following comment: _“But this setup does not really capture the real difficulty of sampling, which usually lies in the amount of computation needed to generate a sample, or the number of queries to the target distribution needed.”_ Our framework is exactly intended to understand the number of queries to the target distribution needed (what we call the sample complexity). Note that for the distribution classes we consider, sample complexity is the bottleneck--our algorithms run in essentially linear time.

---

> > ### Comment · Reviewer_b21p · 2021-08-13
> > **Further response**
> >
> > Thank you for your response. I would first like to apologize that my review had parts that were not clearly phrased, especially the point about the task being reduced to "trying to find a distribution that is close in total variation to the target distribution".
> >
> > Your response did help me better understand your problem set up. As someone who is not an expert in differential privacy, it would have made more sense to me if your setup is motivated with the following problem: given samples from a distribution, can we generate another point that is from the same distribution, but also "unrelated" to the previous samples? This setup arises naturally from generative models, because we want new samples that are different than our training examples. Then one could make the point that differential privacy is actually a natural way to quantify how "unrelated" the new sample is from the training samples, and then study the problem of differentially private sampling.
> >
> > In contrast, the motivation given in the current draft is still unclear to me. Yes, sampling is important, but why should one care about differentially private sampling? The comparison with non-private sampling is confusing: the trivial solution is more of an evidence that the definition of non-private sampling is broken (does not capture the fact that when we sample, we want "unrelated" new data points), rather than a motivation that one should study private sampling. For me, skipping the comparison between non-private and private, and simply mentioning that differential privacy is a natural objective for generative modeling would make a clearer message, and broaden the scope of the contribution.
> >
> > This suggestion is a matter of presentation, rather than of content, so the authors should decide for themselves whether to make this change. But I do feel that such a change gives much better context for the problem, and I would be willing to increase my score of the paper to 6 if it is made.

---

> > > ### Author Response · Authors · 2021-08-16
> > > **motivation**
> > >
> > > Thanks for clarifying the question.
> > >
> > > Our motivation was not really about generating new, unrelated observations. That problem is more explicitly captured by the paper of Axelrod et al (which we cite), where there are no privacy constraints but the goal is to generate new observations that come from the right distribution but are (close to) independent from the ones in the data set.
> > >
> > > Our motivation was understanding what sorts of statistical tasks are possible subject to strong privacy constraints, and whether weakening the goal—to require just sampling instead of learning—changes its difficulty. We hope that our results sheds light on what makes differential privacy challenging. For instance, we show that without a priori information about a distribution on $[k]$, one needs a data set of $n=\Omega(k/\epsilon)$ observations to generate even a single value from the support of the distribution.
> > >
> > > "_For me, skipping the comparison between non-private and private, and simply mentioning that differential privacy is a natural objective for generative modeling would make a clearer message, and broaden the scope of the contribution._"
> > >
> > > That makes sense. We included the discussion of the nonprivate task because we wanted to emphasize how different the private version is. But your point is well taken, and we will think about how to change the flow of the intro to make the problem and motivation more clear. We appreciate the thoughtful feedback.

---

### Official Review · Reviewer_haGa · 2021-07-16

**Rating:** 7
**Confidence:** 4

**Summary:**

This paper initiates an algorithmic study of how to make sampling from discrete distributions private. The formulation is as follows: Given $n$ independent samples from a discrete distribution $P$ over a universe $U$,  output one sample $y$ whose distribution is close to $P$ in total variation distance.  The efficiency goal is to make the number of samples $n$ as small as possible. This appears to be a trivial task - just output any one sample from the input. However, if we require the algorithm to be differentially private then the task is non-trivial and this is the concern of the paper. The paper first considers the problem of private sampling for general distributions over a domain of size $k$ and establishes matching upper and lower bounds on the sample complexity. Then it considers private sampling from product of Bernoulli distributions and establishes upper and lower bounds. In this case, the complexity is not yet completely clear as there is a gap in their bounds.

**Limitations And Societal Impact:**

The authors have sections on both limitations and societal impact which appears adequate.

**Main Review:**

Overall, this is a nice paper initiating an algorithmic investigation of how to sample from distributions with privacy constraints. The paper establishes solid results that should be of interest to a general NeurIPS audience and hence will be a good paper in the conference. The paper is clearly and nicely written.

**Time Spent Reviewing:**

4

---

### Official Review · Reviewer_xLGh · 2021-07-17

**Rating:** 7
**Confidence:** 3

**Summary:**

The paper initiates the study of the following sampling problem under privacy: we are given n samples from a (discrete) distribution P, and want to output a (random) value x such that the distribution of the value we output (over the randomness of the samples and the coins of our algorithm) has small total variation distance (TVD) with P. The problem is trivial normally, since the first sample we are given is distributed exactly according to P. When we are concerned with privacy of the samples, i.e. want to output a value x with approximate differential privacy (viewing the n samples as a database), outputting the first sample is non-private, and the problem is more interesting. The motivation for studying this task, is that it is a weaker version of the problem of learning the distribution P (if we learn an approximation of the distribution, we can clearly approximately sample from it).

The authors consider the problem in three settings: When the support of P is [k], when P is a product distribution on the d-dimensional hypercube (i.e. each bit is independent Bernoulli, not necessarily identically distributed), and the distribution over the hypercube but when each coordinate is a Bernoulli with mean in [1/3, 2/3]. They show lower and upper bounds for each problem when delta = 1/n^c for some constant c > 1. For distributions on [k], the authors show matching upper and lower bounds of needing k/alpha*eps to get a sampler within TVD alpha. For product distributions on hypercubes, they show d/alpha*eps samples suffice, but the lower bound is sqrt(d). The authors are able to match this bound, obtaining TVD alpha with sqrt(d)/epsilon + log (d/alpha) samples when the coordinates are Bernoulli with mean close to 1/2.

In addition to quantitative results, the sample complexities shed some light on the problem of learning P rather than outputting a sample from P. In particular, the best upper bounds for private learning match the sum of upper bounds for nonprivate learning and the upper bounds for private sampling given in this paper. The authors' results also show that e.g. in the Bernoulli/hypercube setting, the regime where sampling is easiest is the regime where learning is hardest and vice-versa.

**Limitations And Societal Impact:**

Yes

**Main Review:**

The paper considers a problem that is new theoretically, but very fundamental. Despite the seeming simplicity of the problem, the results are fairly technical and while some draw on ideas from other papers, I don't believe any part of the analysis simply reduces to reusing techniques from other papers. The distinction between the sampling and learning problems are made very clear in the intro (although some more discussion on what techniques in the paper are fundamentally new, if any, might be useful). Despite the line of work being new, the authors point out that their results also shed more light on and connect together past results in learning, suggesting the problem is new but still interesting, not new for the sake of being new, adequately citing past works of interest.

The work is technically sound and interesting, the analyses feel very complete and did not leave me wondering about details, and the work is complete. The submission is also open about limitations of their work/point out several future directions in this line of work, suggesting this paper could inspire more papers in this area.

The introduction is clearly written and does a good job explaining the qualitative improvements of the paper, not just pointing out the quantitative comparisons with past work. A weakness of the submission is that only one result is covered extensively in the main body, however I view this as more likely a function of the paper length limit rather than poor planning by the authors; the analyses deferred to the appendix are nicely summarized by the author in the introduction. In general I felt like after one pass over the paper I already had a pretty good understanding of the work and its importance.

The main weakness of the paper is that it is not yet clear how important/fundamental this line of work and this paper will be in the future, but the authors do a very good job showing that their results are interesting in relation to other problems in this area, and generally motivating initiating the study of this problem.

**Time Spent Reviewing:**

4

---

### Decision · Program_Chairs · 2021-09-27

**Decision:**

Accept (Poster)

**Comment:**

This paper initiates the study of differentially private *sampling* (as opposed to learning) discrete distributions. The authors obtain nearly tight results for this problem, when the underlying distribution is discrete on a finite alphabet or a binary product distribution. The reviewers agreed that the question studied is interesting and the results are technically novel. Overall, there was a consensus that this paper is above the acceptance threshold.